# Characterization of the Role of Extracellular Vesicles Released from Chicken Tracheal Cells in the Antiviral Responses against Avian Influenza Virus

**DOI:** 10.3390/membranes12010053

**Published:** 2021-12-31

**Authors:** Kelsey O’Dowd, Laura Sánchez, Jennifer Ben Salem, Francis Beaudry, Neda Barjesteh

**Affiliations:** 1Research Group on Infectious Diseases in Production Animals (GREMIP), Department of Pathology and Microbiology, Faculty of Veterinary Medicine, Université de Montréal, Saint-Hyacinthe, QC J2S 2M2, Canada; kelsey.odowd@umontreal.ca; 2Swine and Poultry Infectious Disease Research Center (CRIPA), Faculty of Veterinary Medicine, Université de Montréal, Saint-Hyacinthe, QC J2S 2M2, Canada; Laura.sanchez@umontreal.ca; 3Animal Pharmacology Research Group of Quebec (GREPAQ), Department of Veterinary Medicine, Faculty of Veterinary Medicine, Université de Montréal, Saint-Hyacinthe, QC J2S 2M2, Canada; jennifer.ben.salem@umontreal.ca (J.B.S.); francis.beaudry@umontreal.ca (F.B.); 4Centre de Recherche sur le Cerveau et L’apprentissage (CIRCA), Université de Montréal, Montreal, QC J2S 2M2, Canada

**Keywords:** chicken, proteomics, antiviral responses, avian influenza virus, extracellular vesicles, chicken tracheal cells, macrophages

## Abstract

During viral respiratory infections, the innate antiviral response engages a complex network of cells and coordinates the secretion of key antiviral factors, such as cytokines, which requires high levels of regulation and communication. Extracellular vesicles (EVs) are particles released from cells that contain an array of biomolecules, including lipids, proteins, and RNAs. The contents of EVs can be influenced by viral infections and may play a role in the regulation of antiviral responses. We hypothesized that the contents of EVs released from chicken tracheal cells are influenced by viral infection and that these EVs regulate the function of other immune cells, such as macrophages. To this end, we characterized the protein profile of EVs during avian influenza virus (AIV) infection and evaluated the impact of EV stimulation on chicken macrophage functions. A total of 140 differentially expressed proteins were identified upon stimulation with various stimuli. These proteins were shown to be involved in immune responses and cell signaling pathways. In addition, we demonstrated that EVs can activate macrophages. These results suggest that EVs play a role in the induction and modulation of antiviral responses during viral respiratory infections in chickens.

## 1. Introduction

During viral respiratory infections, host innate responses aim to prevent viral entry and replication through a variety of strategies, which collectively act as the first line of defense prior to the induction of adaptive immune responses. Upon the infection with avian viral respiratory pathogens, such as avian influenza virus (AIV), epithelial cells become the primary target of the virus [1,2,3]. Innate antiviral responses involve a complex network of cells, including macrophages and dendritic cells, which can engage cell sensors, such as pattern recognition receptors (PRRs), that detect viral components and activate specific signaling pathways. Furthermore, AIV can infect macrophages [2]. Activation of epithelial cells and macrophages following pathogen recognition leads to the recruitment of other cells of the immune system and subsequent production of interferons (IFNs), interleukins (ILs), and other pro-inflammatory cytokines. Moreover, type I IFNs, IFN-α and IFN-β, induce an antiviral state in virus-infected cells and in neighboring cells by initiating the production of IFN-stimulated genes (ISGs), which can interfere with the viral replication cycle and contribute to pathogen clearance [2,3,4]. In addition, macrophages have other functions essential for host defense against pathogens, such as phagocytosis, secretion of antimicrobial peptides, and antigen presentation [5]. 

Given the complex and highly regulated immune responses and the extensive host-pathogen interactions involved during infection, regulated communication and coordination between host cells are essential for efficient detection, regulation, and clearance of invading pathogens. This communication can be through the production of cytokines, direct cell-to-cell contact, or the secretion of extracellular vesicles (EVs) [6]. EVs are a heterogenous group of lipid-encapsulated particles released by all cell types and measuring 30–1000 nm [7]. Several subcategories of EVs, such as exosomes, microvesicles, and apoptotic bodies, are characterized based on biogenesis pathways, size, and specific protein markers [8,9]. Following secretion, EVs can be taken up through several mechanisms, including endocytosis, phagocytosis, or membrane fusion [10]. These particles contain a diverse cargo of biomolecules, including lipids, proteins, and ribonucleic acids (RNAs), such as microribonucleic acids (miRNAs), which can play a role in information transfer and act as a cellular “language” [8,11,12]. These regulators of gene expression have been shown to regulate several biological processes, including immune responses, which makes them key molecules of interest for EV profiling studies [13]. In addition, several studies have shown that viral infections can affect EV contents and influence information transfer and resulting immune responses [14,15,16,17,18].

The contents and specific regulatory roles of EVs are poorly described in the context of infectious diseases in chickens. The few studies that have characterized the contents of EVs in the context of immune responses in chickens focus primarily on miRNA profiles, with little focus on protein content; therefore, studies are required to evaluate the role of EV protein content in the induction and modulation in the host response to viral infections [19,20,21,22,23,24,25,26]. For example, a recent study evaluated the proteomic profile of serum exosomes from Marek’s Disease Virus (MDV)-vaccinated and protected and lymphoma-bearing chickens and identified potential biomarkers for the disease [24]. The characterization of EV proteomic profiles will allow for a better understanding of EV dynamics and important insight for the development of new strategies for the control of viral infections in chickens using EVs.

Furthermore, studies evaluating the impact of EV stimulation on immune cell functions in chickens are rare. In macrophages specifically, there are two studies assessing the impact of stimulating chicken macrophages with macrophage-derived exosomes stimulated with either polyinosinic: polycytidylic acid (polyI:C) or lipopolysaccharide (LPS). These studies revealed that these exosomes can modulate the immune response through specific antiviral pathways, such as the NF-κB signaling pathway [27,28].

We previously described the induction of antiviral responses and communication between tracheal cells and macrophages in the chicken respiratory system [29,30,31,32]. Furthermore, we showed that chicken tracheal cells release EVs. Moreover, we found that the miRNA profile of EVs is influenced by the type of stimuli [33]. In this study, we aimed to investigate another component of the EV cargo and characterize the protein content of EVs released from chicken tracheal cells and their functional properties, including their impact on the function of macrophages. We hypothesized that viral infection influences the contents of EVs released from chicken tracheal cells and that these EVs regulate the function of other immune cells, such as macrophages. Ultimately, we characterized the protein profile of EVs and evaluated the impact of EVs on cells of the immune system in the context of antiviral responses against avian influenza virus infection.

## 2. Material and Methods

### 2.1. Avian Influenza Virus (AIV)

Ten-day-old specific-pathogen-free (SPF) embryonated chicken eggs (layer chickens, white Leghorn, Canadian Food Inspection Agency, Ottawa, ON, Canada) were used to propagate the low pathogenic avian influenza virus AIV A/Duck/Czech/56 (H4N6) by inoculation through the allantoic cavity [34]. Briefly, the eggs were candled to verify proper embryo development and 100 µL of stock allantoic fluid containing 0.2 hemagglutinin units (HAU) of the H4N6 virus was injected through the allantoic cavity. The allantoic fluid was harvested from the eggs at 48 h post-inoculation and the virus titer was determined using endpoint dilution in Madin-Darby Canine Kidney (MDCK) cells (a gracious gift from Dr. Shayan Sharif’s laboratory at the Ontario Veterinary College, University of Guelph, ON, Canada) [35].

### 2.2. Toll-Like Receptor (TLR) Ligands

TLR ligands lipopolysaccharide (LPS) from *Escherichia coli* 026:B6 (Sigma-Aldrich, Oakville, ON, Canada) and polyinosinic:polycytidylic acid (polyI:C) (InvivoGen, San Diego, CA, USA) were used in this study. These TLR ligands were selected because they were previously shown to induce immune responses in chicken tracheal cells [31,32].

### 2.3. Tracheal Organ Culture (TOC)

TOC was performed as previously described [31]. Briefly, tracheas were aseptically collected from nineteen-day-old SPF chicken embryos (Canadian Food Inspection Agency, Ottawa, ON, Canada) and washed twice with warm Hanks’ balanced salt solution (HBSS, Gibco, Burlington, ON, Canada) to remove excess mucus. The connective tissues surrounding the trachea were removed by thorough dissection. Tracheas were then manually dissected into 1 mm rings using razor blades. The rings were transferred into 6-well cell culture plates (one embryo per well, 10–15 rings per embryo) containing phenol red-free complete Medium 199 (Sigma–Aldrich, Oakville, ON, Canada) supplemented with 10% EV-depleted and heat-inactivated fetal bovine serum (FBS, Gibco, Burlington, ON, Canada), 2 mM GlutaMax supplement (Gibco, Burlington, ON, Canada), 25 mM 4-(2-hydroxyethyl)-1-piperazineethanesulfonic acid (HEPES) buffer (Gibco, Burlington, ON, Canada), 100 U/mL penicillin/100 µg/mL streptomycin (Gibco, Burlington, ON, Canada), and 50 µg/mL gentamicin (Gibco, Burlington, ON, Canada). To prepare EV-depleted FBS, FBS was first heat-inactivated at 56 °C for 30 min and then complete Medium 199 containing 20% FBS was ultracentrifuged at 100,000× *g* for 18 h (4 °C) (38.5 mL, Open-Top Thinwall Ultra-Clear Tube, 25 × 89 mm 344,058 and Optima L-100XP, Beckman Coulter, Mississauga, ON, Canada). The supernatant was then collected and filtered through a 0.2 μm syringe filter (VWR, Montreal, QC, Canada) and diluted with complete Medium 199 to reach a final concentration of 10% FBS.

The tracheal rings were incubated at room temperature on a low-speed benchtop rocker for three hours to exclude mucus production and potential reactions from the process of TOC preparation. Following the incubation, the media was replaced with fresh complete Medium 199 containing 10% FBS. During the experiments, the ciliary activity of the TOC was monitored and confirmed under a light microscope.

### 2.4. Chicken Macrophage Cell Line

The Muquarrab Qureshi-North Carolina State University (MQ-NCSU) cell line, a gracious gift from Dr. Shayan Sharif’s laboratory at the Ontario Veterinary College, University of Guelph, ON, Canada, is an avian macrophage cell line derived from spleen cells infected with the JM/102W strain of Marek’s disease virus [36]. The MQ-NSCU cells were cultured in LM-HAHN media composed of a 1:1 ratio of McCoy’s 5 A (modified) medium (Gibco, Burlington, ON, Canada) and Leibovitz’s L-15 medium (Gibco, Burlington, ON, Canada) supplemented with 8% heat-inactivated FBS (Gibco, Burlington, ON, Canada), 10% heat-inactivated chicken serum (Gibco, Burlington, ON, Canada), 1% tryptose phosphate broth (Gibco, Burlington, ON, Canada), 1% sodium pyruvate (Gibco, Burlington, ON, Canada), 2 mM GlutaMax supplement (Gibco, Burlington, ON, Canada), 100 U/mL penicillin/100 µg/mL streptomycin (Gibco, Burlington, ON, Canada), and 50 μg/mL gentamicin (Gibco, Burlington, ON, Canada) at 40 °C and 5% CO_2_ in a humidified incubator.

### 2.5. Determining Protein Content of EVs Released from TOC

#### 2.5.1. TOC Infection with AIV and Stimulation with TLR Ligands

For AIV infection of TOCs, tracheal rings were infected with 10^4^ pfu/mL. For the TLR ligand stimulation of TOCs, tracheal rings were stimulated with either LPS (1 µg/mL) or polyI:C (25 µg/mL) (doses were selected based on previous studies in chickens) [29,31,32]. TOCs were incubated at 40 °C and 5% CO_2_ in a humidified incubator. Infection/stimulation for all treatment groups was done in complete FBS-free Medium 199 as animal sera contain non-specific inhibitors of influenza viruses [37]. Furthermore, the control groups received complete FBS-free Medium 199. At 2 h post-stimulation/-infection, tracheal rings were washed twice with HBSS (Gibco, Burlington, ON, Canada) before incubation at 40 °C and 5% CO_2_ in a humidified incubator in fresh complete FBS-free Medium 199.

#### 2.5.2. EV Isolation

After a 24 h incubation period at 40° C and 5% CO_2_ in a humidified incubator, TOC supernatants were collected. There were two replicates per treatment group; each replicate consisting of supernatants pooled from three wells (three individual embryos). An optimized ultracentrifugation protocol for EV isolation from TOC supernatants was used [33]. Briefly, all the centrifugation and ultracentrifugation was performed at 4 °C. Ultracentrifugation was performed using the 17 mL, Polypropylene Tube, 16 × 96 mm, and Optima L-100XP (Beckman Coulter, Mississauga, ON, Canada). Supernatants were first centrifuged at 300× *g* for 10 min to remove cellular debris. Supernatants were then recovered and centrifuged at 2000× *g* for 20 min. Supernatants were again recovered and ultracentrifuged at 10,000× *g* for 30 min. Supernatants were recovered and filtered with 0.2 µm syringe filters (VWR, Montreal, QC, Canada), followed by ultracentrifugation at 100,000× *g* for 60 min. Supernatants were discarded and pellets were resuspended in FBS-free complete Medium 199 and ultracentrifuged at 100,000× *g* for a final 60 min. Following the final round of ultracentrifugation, the supernatants were discarded. For samples designated for mass spectrometry (MS) analysis, the pellets were resuspended in 50 µL of 50 mM ammonium bicarbonate (pH = 8) (Sigma-Aldrich, Oakville, ON, Canada) and stored at −80 °C. Samples designated for protein concentration determination and macrophage experiments were resuspended in phosphate-buffered saline (PBS, Gibco, Burlington, ON, Canada). The protein concentrations of the isolated EVs were determined using the Micro BCA Protein Assay Kit according to the manufacturer’s instructions (Thermo Fisher Scientific, Burlington, ON, Canada). To validate the purity of the EVs isolated using this protocol, the presence of specific EV protein markers and the morphology of EVs were confirmed by Western Blot and transmission electron microscopy (TEM), respectively, in our previous study [33].

#### 2.5.3. Sample Preparation for Mass Spectrometry Analysis

To identify the proteins present in the EV samples by MS, protein digestion was performed using the In-Solution Tryptic Digestion according to the manufacturer instructions (Thermo Fisher Scientific, Burlington, ON, Canada) with some modifications. Briefly, the samples were thawed and an additional 50 µL of 50 mM ammonium bicarbonate (pH = 8.0) (Sigma-Aldrich, Oakville, ON, Canada) including a cocktail of proteinase inhibitors (cOmplete, Mini Protease Inhibitor Cocktail, Sigma-Aldrich, Oakville, ON, Canada) was added. Samples were homogenized by bead mill homogenization using reinforced 1.5 mL homogenizer tubes containing 50 mg glass beads. The samples were homogenized with three bursts of 60 s at a speed of 5 m/s. Proteins were precipitated by adding cold acetone at a ratio of 1/5 (*v*/*v*). Samples were then centrifuged at 12,000× *g* for 10 min, supernatants were discarded, and pellets were resuspended in 50 mM tris hydrochloride (Tris-HCl) buffer (pH = 8.0). Denaturation of proteins was done at 95 °C for 15 min and allowed to cool. Reduction and alkylation were performed as follows: samples were reduced with 20 mM dithiothreitol (DTT) at 90 °C for 15 min and alkylated with 40 mM iodoacetamide (IAA) at room temperature for 30 min protected from light. The alkylation reaction was quenched with the addition of DTT (10 mM final concentration). Five micrograms of proteomic-grade trypsin was added. The reaction was performed at 37 °C for 24 h. Finally, the protein digestion was quenched by adding 10 µL of a 1% trifluoroacetic acid (TFA) solution. Samples were centrifuged at 12,000× g for 10 min, and the supernatants were transferred into injection vials for analysis.

#### 2.5.4. Mass Spectrometry-Based Proteomics

The analyses were carried out on a Vanquish FLEX Ultra High-Performance Liquid Chromatography (UHPLC) system coupled to a Q Exactive Plus Orbitrap Mass Spectrometer (Thermo Scientific, San Jose, CA, USA). HPLC separation was performed using gradient elution with a microbore column Thermo Biobasic C18 150 × 1 mm, with a particle size of 5 μm. The 5 µL of sample was separated at a flow rate of 50 µL/min using a gradient elution strategy. The initial mobile phase condition consisted of acetonitrile and water (both fortified with 0.1% of formic acid) at a ratio of 5:95. From 0 to 3 min, the ratio was maintained at 5:95. From 3 to 123 min, a linear gradient was applied up to a ratio of 40:60 and maintained for 3 min. The mobile phase composition ratio was reverted at the initial conditions and the column was allowed to re-equilibrate for 25 min. The Q Exactive Plus Orbitrap Mass Spectrometer was interfaced with the UHPLC system using a pneumatic assisted heated electrospray ion source. Nitrogen was used for sheath and auxiliary gases and was set at 10 and 5 arbitrary units. Auxiliary gas was heated to 200 °C. The heated electrospray ionization (ESI) probe was set to 4000 V and the ion transfer tube temperature was set to 300 °C. MS detection was performed in positive ion mode and operating in TOP-10 Data Dependent Acquisition (DDA). A DDA cycle entailed one MS^1^ survey scan (*m/z* 400–1500) acquired at 70,000 resolution (FWHM) and precursors ions meeting user-defined criteria for charge state (i.e., *z* = 2, 3, or 4), monoisotopic precursor intensity (dynamic acquisition of MS^2^ based TOP-10 most intense ions with a minimum 1 × 10^4^ intensity threshold). Precursor ions were isolated using the quadrupole (1.5 Da isolation width) and activated by HCD (28 NCE), and fragment ions were detected in the Orbitrap at 17,500 resolution (FWHM). Datasets were analyzed using Thermo Proteome Discoverer (version 2.4) in combination with SEQUEST using default settings unless otherwise specified. SEQUEST used a curated database consisting of FASTA sequences extracted from UniProt (*Gallus* reference proteome, proteome identifier UP000000539). Parameters were set as follows: MS^1^ tolerance of 10 ppm; MS^2^ mass tolerance of 0.02 Da for Orbitrap detection; enzyme specificity was set as trypsin with two missed cleavages allowed; carbamidomethylation of cysteine was set as a fixed modification; and oxidation of methionine was set as a variable modification. The minimum peptide length was set to six amino acids, and proteins identified by only one peptide were removed. Datasets were further analyzed with Percolator to improve the rate of confident peptide identifications [38]. Peptide-spectrum-matches (PSMs) and protein identification were filtered at 1% false discovery rate (FDR) threshold. For protein quantification and comparative analysis, we used the peak integration feature of the Proteome Discoverer 2.4 software [39]. For each identified protein, the average ion intensity of unique peptides was used for protein abundance.

#### 2.5.5. Protein Functional Analyses

The abundance ratios were generated from (Sample)/(Control) abundance values. Volcano plots were generated for proteins that had an associated abundance ratio and *p*-value (Benjamini-Hochberg method). Proteins with an abundance ratio ≥2-fold change and a *p*-value < 0.05 were considered differentially expressed (DE) and retained for downstream analysis. The databases ExoCarto (http://www.exocarta.org/; accessed date: 16 August 2021) and Vesiclepedia (http://microvesicles.org/index.html#; accessed date: 16 August 2021) were used to screen for exosome-associated proteins [40,41]. Venn diagram analysis for the DE proteins among the different treatment groups was performed using the online tool http://bioinformatics.pbs.ugent.be/webtools/Venn/ (accessed date: 17 November 2020). Functional annotation of the DE proteins was performed by gene ontology (GO) mapping using PANTHER (Protein Analysis Through Evolutionary Relationships Classification System, http://pantherdb.org/; accessed date: 19 August 2021) [42]. In addition, the associated pathways extracted from the PANTHER database were used to create tables and networks, which were then imported into Cytoscape to generate diagrams representing these networks [43]. Furthermore, the STRING database (Search Tool for the Retrieval of Interacting Genes/Proteins, https://string-db.org/; accessed date: 20 December 2021) was used to assess and illustrate relationships among the DE proteins (medium confidence score of 0.400) [44].

### 2.6. Treatment of Chicken Macrophages with EVs Released from TOC

#### 2.6.1. EV Uptake by Chicken Macrophages

MQ-NCSU cells were seeded in 24-well plates in DMEM cell culture media composed of *Dulbecco’s Modified Eagle Medium* (*DMEM*, Gibco, Burlington, ON, Canada) supplemented with 10% FBS (Gibco, Burlington, ON, Canada), 2 mM GlutaMax Supplement (Gibco, Burlington, ON, Canada), 100 U/mL penicillin/100 µg/mL streptomycin (Gibco, Burlington, ON, Canada), and 50 μg/mL gentamicin (Gibco, Burlington, ON, Canada) at a cell density of 5 × 10^4^ cells/well and incubated at 40 °C and 5% CO_2_ in a humidified incubator overnight. The cells were then labeled with PKH67 Green Fluorescent Cell Linker Kit for General Cell Membrane Labeling (Sigma-Aldrich, Oakville, ON, Canada) according to the manufacturer’s instructions with modifications. Briefly, 4 μL of PKH67 dye was added to 1 mL of Diluent C and 100 μL of the solution was added to each well. Following a 5-min incubation period at room temperature, 100 μL of 1% BSA (Gibco, Burlington, ON, Canada) was added to each well for 1 min. Finally, cells were washed three times with PBS. Unlabeled macrophage controls were also included. EVs isolated from TOC were labeled with PKH26 Red Fluorescent Cell Linker Kit for General Cell Membrane Labeling (Sigma-Aldrich, Oakville, ON, Canada) according to the manufacturer’s instructions with some modifications, as previously described [45]. Briefly, the EVs were added to 1 mL Diluent C, and 4 μL of PKH26 dye was added to 1 mL Diluent C. The EVs and dye were mixed and incubated at room temperature for 5 min before 2 mL of 1% BSA (Gibco, Burlington, ON, Canada) was added. PBS-PKH26 controls were included. Using Amicon Ultra-4 Centrifugal Filter Unit Ultracel 100 k (MilliporeSigma, Burlington, MA, USA), the samples were centrifuged at 4000× *g* and then washed three times with PBS, before being washed twice with DMEM (Gibco, Burlington, ON, Canada). Following the labeling and incubation, MQ-NCSU cells were treated with 10 µg labeled EVs per well and incubated for 2 h. Untreated PKH67-labelled macrophage controls were included. Cells were then washed twice with PBS (Gibco, Burlington, ON, Canada) and fixed in 4% formalin (Sigma-Aldrich, Oakville, ON, Canada) for 15 min at room temperature. Finally, cells were washed twice with PBS (Gibco, Burlington, ON, Canada). EV uptake by macrophages was then visualized using a Leica DMI 4000B automated inverted fluorescence microscope with a Leica DFC 490 digital camera and the Leica Application Suite Software, version 3.8.0 (Leica Microsystems Inc., Richmond Hill, ON, Canada). Fiji/ImageJ software and the stitching plugin were used for image analysis [46,47].

#### 2.6.2. Nitric Oxide (NO) Production by Chicken Macrophages Treated with EVs

MQ-NCSU cells were seeded in 48-well plates at a viable cell density of 5 × 10^5^ cells/well in DMEM cell culture media at 40 °C and 5% CO_2_ in a humidified incubator for 2 h. The cells were then stimulated with two different doses of EVs isolated from TOC: low (5 µg/mL) and high (50 µg/mL). Furthermore, for LPS-stimulated treatment groups, cells were stimulated with 1 µg/mL LPS at 1 h post-stimulation with EVs. Untreated and LPS-stimulated macrophage controls were included. The ability of EVs to stimulate nitric oxide production in culture supernatants collected 48 h post-stimulation was evaluated by the Griess assay method. Briefly, 50 µL of sulfanilamide solution (1% sulfanilamide and 5% phosphoric acid in water) (sulfanilamide, Thermo Fisher Scientific, Burlington, ON, Canada) and phosphoric acid, Sigma-Aldrich, Oakville, ON, Canada) were added to 50 µL of samples in wells of a microplate and incubated for 10 min at room temperature, protected from light. Next, 50 µL of NED solution (0.1% N-1-napthylethylenediamine dihydrochloride in water) (N-(1-Naphthyl)-ethylenediamine dihydrochloride, Bio Basic, Markham, ON, Canada) was added to each well and the plate was then incubated for 10 min at room temperature, protected from light. Finally, absorbance was measured using a plate reader with a 540 nm filter. A nitrite standard (sodium nitrite, Bio Basic, Markham, ON, Canada) with a linear range of 0–100 µM was included to generate a reference curve.

#### 2.6.3. Phagocytosis by Chicken Macrophages Treated with EVs

MQ-NCSU cells were seeded in 96-well Black Polystyrene Microplates (Corning, NY, USA) at a viable cell density of 7.5 × 10^4^ cells/well in DMEM cell culture media and incubated at 40 °C and 5% CO_2_ in a humidified incubator for 2 h. The cells were then stimulated with two different doses of EVs isolated from TOC: low (5 µg/mL) and high (25 µg/mL) and incubated at 40 °C and 5% CO_2_ in a humidified incubator. At 12 h post-stimulation with EVs, phagocytosis was assessed using pHrodo Red *Escherichia coli* Bioparticles Conjugates for Phagocytosis (Invitrogen, Burlington, ON, Canada) according to the manufacturer’s instructions, with some modifications. Briefly, pHrodo Red *Escherichia coli* Bioparticles Conjugates were resuspended in 2 mL Live Cell Imaging Solution (Invitrogen, Burlington, ON, Canada) and sonicated to homogeneously disperse the particles. For each well, 25 µL of the medium was removed and replaced with 25 µL of resuspended beads. Untreated macrophage controls were included. The samples were then incubated at 37 °C for 4 h. Finally, fluorescence was evaluated using a fluorescence plate reader at an excitation/emission spectra of 560/586 nm.

#### 2.6.4. Gene Expression of Chicken Macrophages Treated with EVs

MQ-NCSU cells were seeded in 48-well plates at a viable cell density of 5 × 10^5^ cells/well in DMEM cell culture media at 40 °C and 5% CO_2_ in a humidified incubator for 2 h. The cells were then stimulated with two different doses of EVs isolated from TOC: low (5 µg/mL) and high (25 µg/mL). Untreated macrophage controls were included. At 3 h or 18 h post-stimulation with EVs, cells were collected in TRIzol reagent (Invitrogen, Burlington, ON, Canada) and total RNA was extracted according to the manufacturer’s instructions. Total RNA was treated with the DNA-free DNase Kit (Ambion, Austin, TX, USA) according to manufacturer’s instructions. For each sample, complementary DNA (cDNA) synthesis was performed with 1 µg of RNA using the Maxima Reverse Transcriptase kit (Thermo Scientific, Burlington, ON, Canada) according to the manufacturer instructions and using Oligo(dT)_20_ primer (Invitrogen, Burlington, ON, Canada), dNTP Mix (Thermo Scientific, Burlington, ON, Canada), and RNAseOUT (Invitrogen, Burlington, ON, Canada). Quantitative real-time polymerase chain reaction (RT-PCR) was carried out with cDNA diluted 1:10 in DEPC-treated water and PowerUp SYBR Green Master Mix (Thermo Scientific, Burlington, ON, Canada) according to the manufacturer instructions using the ViiA 7 Real-Time PCR system (Applied Biosystems, Waltham, MA, USA), as previously described [29,30,32]. Briefly, the cycling program consisted of 50 °C for 2 min, 95 °C for 10 min, following by 45 cycles of 95 °C for 10 s, 60 °C or 64 °C annealing for 5 s, depending on the specific primer (Table 1), and elongation and signal acquisition at 72 °C for 10 s. Melting curve analysis was done in three steps: 95 °C for 15 s, cooling to 60 °C for 1 min, and heating to 95 °C for 15 s. Finally, data analysis to calculate the relative gene expression was done using the Pfaffl method [48].

### 2.7. Statistical Analysis

Statistical analysis of the NO production and phagocytosis by chicken macrophages treated with EVs data was performed by one-way ANOVA followed by the Tukey test for multiple comparisons. Statistical analysis of the gene expression of chicken macrophages treated with EVs data was performed by student’s *t*-test (two groups). For all statistical analyses, calculations were performed using GraphPad Prism software version 9.2.0 (La Jolla, CA, USA) and *p*-value < 0.05 was considered statistically significant.

## 3. Results

### 3.1. EVs Released from TOCs Treated with AIV, LPS, and PolyI:C Have Distinctive Protein Profiles

The protein profiles were evaluated to determine the ability of AIV infection and LPS and polyI:C stimulation to influence the protein contents of EVs. A total of 140 known DE proteins were identified among all treatment groups, with 52 up-regulated and 88 down-regulated proteins (Table 2 and Table 3). Using the associated gene symbols, we then referred to the vesicle proteome databases ExoCarta and Vesiclepedia, which contain records of proteins previously shown to be EV-associated [40,41]. Among the DE proteins, 59 were identified in the Vesiclepedia database only, whereas 52 were found in both the ExoCarta and Vesiclepedia databases, for a total of 111, representing just under 80% of our identified proteins (Appendix A). In addition, 29 proteins not previously found in EVs were identified. It is also important to note that no AIV proteins were found within EVs from our AIV treatment group. Furthermore, the EV protein marker *Homo sapiens* Tumor susceptibility gene 101 (TSG101) was identified in the EVs [9]. Among the AIV, LPS, and polyI:C treatment groups, a total of 67, 85, and 76 DE proteins were identified, respectively. Proteins were considered DE if they satisfied the *p*-value < 0.05 and abundance ratio ≥ 2-fold change threshold conditions. Furthermore, 31, 31, and 24 proteins were up-regulated, whereas 36, 54, and 52 proteins were down-regulated in the AIV, LPS, and polyI:C treatment groups, respectively (Figure 1). Several of the proteins identified in the treated groups were found in more than one treatment group (Figure 2 and Appendix A). In all treatment groups, 12 common proteins were up-regulated, and 17 common proteins were down-regulated. In addition, within the AIV, LPS, and polyI:C groups, 9, 17, and 4 proteins were uniquely up-regulated, and 9, 22, and 20 proteins were uniquely down-regulated, respectively. Furthermore, four proteins presented different expression patterns among the different treatment groups. The protein EXOC5 (F1NF87) was up-regulated in the LPS group, but down-regulated in the AIV group. The COL1A2 (A0A5H1ZRJ7) was up-regulated in the LPS group, but down-regulated in the polyI:C group. The GPD2 (F1NCA2) was up-regulated in the LPS group, but down-regulated in the polyI:C group. The LOC107055115 (A0A1D5P3H2) was up-regulated in the polyI:C group, but down-regulated in the AIV group.

### 3.2. Proteins Found in EVs Released from TOCs Have Functions in Cell Signaling and Immune System Processes

Following differential expression filtering of the proteins, GO analysis was performed to identify the proteome component characteristics. The associated terms for biological process, molecular function, cellular component, protein class, and pathways were determined using the PANTHER database [42]. Overall, the GO analysis characterized common and differing characteristics among the different EV treatment groups (Figure 3 and Appendix A). Concerning molecular function terms, all groups had a high proportion of up- and down-regulated proteins associated with the general terms “catalytic activity” and “binding” (Figure 3a,b). For analysis of biological process terms, up-regulated proteins were associated with categories such as “signaling” and “response to stimulus” terms (Figure 3c). More specifically, the proteins in associated to “signaling” were ADGRA1/GPR123 (A0A3Q2U363) (common to all three treatment groups), IGF1R (A0A1D5PVH7) (AIV group), LATS1 (E1C371) (LPS group), and PKN2 (E1BYS6) (polyI:C group). The proteins associated with “response to stimulus” were ADGRA1/GPR123 (A0A3Q2U363) (common to all three treatment groups), USP53 (E1BSS2) (common to AIV and polyI:C groups), LATS1 (E1C371) (LPS group), and PKN2 (E1BYS6) (polyI:C group). In contrast, down-regulated proteins were associated with “signaling” and “response to stimulus” as well, but also with the term “immune system process” (Figure 3d). The proteins associated with “signaling” were PLCE1 (A0A1D5PQ57) (common to AIV and polyI:C groups), RHBDF2 (A0A3Q2U888) (common to LPS and polyI:C groups), MAGI1 (A0A1D5NU15) (AIV group), WISP1 (A0A1L1RKD5), COL27A1 (F1NHH4), DENND4A (A0A1D5PE26), FYB (E1C908), GRIN2C (R4GFN5), and ZYX (A0A3Q2UIH4) (LPS group), and DENND4C (F1NQ24), STAG1 (A0A1D5NY78), EPHA5 (A0A1D5PES4), and AMHR2/Gga.10225 (E1BQF4) (polyI:C group). The proteins associated to “response to stimulus” were EOMES (R4GH67) (common to all three treatment groups), PLCE1 (A0A1D5PQ57) (common to AIV and polyI:C groups), NCOA1 (Q5F393) and RHBDF2 (A0A3Q2U888) (common to LPS and polyI:C groups), MAGI1 (A0A1D5NU15) (AIV group), TICRR (E1BU88), WISP1 (A0A1L1RKD5), COL27A1 (F1NHH4), DENND4A (A0A1D5PE26), FYB (E1C908), GRIN2C (R4GFN5), TIGAR (R4GIZ6), and ZYX (A0A3Q2UIH4) (LPS group), and DENND4C (F1NQ24), STAG1 (A0A1D5NY78), EPHA5 (A0A1D5PES4), and AMHR2/Gga.10225 (E1BQF4) (polyI:C group).The proteins associated to “immune system process” were EOMES (R4GH67) (common to all three treatment groups) and FYB (E1C908) (LPS group). Furthermore, the proportions for cellular component categories for both and up- and down-regulated proteins of all groups were similar, with the highest proportion of percent gene hits being for “cellular anatomical complex”, followed by “intracellular” and, finally, “protein-containing complex” (Figure 3e,f). Finally, concerning protein component terms, the category with one of the highest percent of gene hits for up- and down-regulated proteins was gene-specific transcriptional regulators (Figure 3g,h).

To gain a better understanding of the role the identified DE proteins, functional annotation for pathways were obtained from the PANTHER classification system database and used to build a network (Figure 4 and Appendix A). In the AIV treatment group, down-regulated proteins PLCE1 (A0A1D5PQ57) and RGS14 (A0A1D5PPP1) were found to be associated with the inflammation mediated by chemokine and cytokine signaling pathways, and down-regulated protein WRN (A0A1L1RQF9) was found to be associated with the p53 pathway (Figure 4a). In the LPS treatment group, up-regulated proteins COL4A1 (A0A1D5P8P3) and COL27A1 (F1NHH4) and down-regulated protein COL1A2 (A0A5H1ZRJ7) were found to be associated with the integrin signaling pathways (Figure 4b). In the polyI:C treatment group, the inflammation mediated by chemokine and cytokine signaling pathway-associated down-regulated protein PLCE1 (A0A1D5PQ57) was also identified. Furthermore, up-regulated protein AMHR2/Gga.10225 (E1BQF4) was found to be associated with the TGF-beta signaling pathway, and the up-regulated protein COL1A2 (A0A5H1ZRJ7) was found to be associated with the integrin signaling pathways. Finally, the up-regulated protein CDH9 (E1C264) was found to be associated with the cadherin signaling and the Wnt signaling pathways (Figure 4c). Moreover, to illustrate and increase the overall understanding of the relationships between the identified DE proteins, protein-protein interaction networks were built using the STRING database (Appendix A). The results of this analysis demonstrate a connected network of proteins within the different treatment groups. For example, in the AIV up-regulated treatment group, we found interactions between the proteins SSPO (Q2PC93), ADGRA1/GPR123 (A0A3Q2U363), PAPLN (A0A1D5P8Q3), FRAS1 (F1NX10), and ARID5B (A0A3Q2UIP4). In the LPS down-regulated group, we found interactions between KAT6A (F1NT94), RHOGL (A0A3Q2TZW2), EPC1 (E1C5B4), and EHMT1 (A0A1D5PVY2), as well as between ANK2 (A0A1D5P124) and SPTBN5 (F1NV58).

### 3.3. EVs Released from TOCs Impact Chicken Macrophage Functions

To determine whether chicken macrophages can uptake EVs, fluorescent staining was executed. EVs labeled with the red fluorescent membrane dye PKH26 were added to macrophages labeled with the green fluorescent dye PKH67. EV uptake was visualized by fluorescence microscopy (Figure 5). We observed EV internalization by macrophages at 2 h post-treatment.

The ability of EVs released from TOC to induce nitric oxide production in macrophage cell culture supernatants 48 h post-stimulation was evaluated (Figure 6). Macrophages were treated with two different doses of EVs only, or with EVs and LPS. For macrophages treated with EVs only, the high dose of EVs for all EV groups, except the EV AIV group, was able to induce a significant increase in NO production compared to the untreated control group. However, there was no significant difference between control group and low dose treatments. Furthermore, a significant difference between EV LPS, EV polyI:C, and EV CTRL low doses and high doses were observed within the respective groups. For the macrophages treated with EVs and 1 µg/mL LPS, a synergistic effect was observed for NO production in comparison to the 1 µg/mL LPS control group. For macrophages treated with EVs and LPS, the high dose of all groups was able to induce a significant increase in NO production compared to the LPS control group. In addition, for the EVs + LPS groups, a significant difference between the doses within each EV group was observed, with the low dose of EVs + LPS groups inducing a higher concentration nitrite of than the high dose of EVs + LPS groups.

Phagocytosis was then evaluated to investigate the effect of EV stimulation on macrophage functions. Phagocytosis was assessed by a fluorescent bead-based assay. Although there appears to be an increase in fluorescence intensity (relative fluorescence units, RFU) in all the EV groups compared to the untreated control, only the high dose of EV LPS group showed a significant increase in fluorescence intensity (Figure 7). Furthermore, no significant differences were observed between low and high doses within each EV group.

To assess the ability of EV treatment to influence the relative gene expression levels in chicken macrophages, macrophages were treated with a low dose (5 µg/mL) or a high dose (25 µg/mL) of EVs and collected at two different timepoints, 3 h and 18 h post-stimulation (Figure 8). The relative expression levels of IFN-α, IFN-β, IL-1β, PKR, and MDA5 were evaluated. We evaluated the relative gene expression levels of macrophages treated with EVs from chicken tracheal cells stimulated with AIV and LPS, as well as our EV CTRL group. Given the large amount of treatment groups from the different dosages and timepoints, we chose to exclude the EV polyI:C for this part of the experiment. The stimulation of macrophages with EV AIV, both low and high doses, as well as with a high dose of EV CTRL induced significant up-regulation of IFN-α at 3 h post-stimulation (Figure 8a). Furthermore, there is an overall decrease in relative gene expression for both IFN-α and IFN-β in all treatment groups at 18 h versus 3 h post-stimulation, with significant differences observed for the high dose of EV AIV for both IFN-α and IFN-β and the low dose of EV LPS for IFN-α only (Figure 8a,b). All treatment groups for both timepoints showed significant upregulation of IL-1β (Figure 8c). In addition, the upregulation of IL-1β for EV AIV and EV LPS groups is dose dependent. For PKR, there was no significant difference in relative gene expression at 3 h post-stimulation; however, there was significant downregulation for all of the EV groups at 18 h post-stimulation (Figure 8d). Similarly, there was no significant difference in relative gene expression at 3 h post-stimulation for MDA5 (Figure 8e). At 18 h post-stimulation, there was significant downregulation for all of the EV groups.

## 4. Discussion

We previously described the induction and regulation of antiviral responses in tracheal cells and macrophages in the chicken respiratory system [29,30,31,32]. To further investigate the mechanisms by which this modulation occurs, we looked at the role of cellular and EV miRNAs in the context of antiviral responses in chicken tracheal cells. We showed the EV miRNA content can be influenced by AIV infection and TLR ligand stimulation [33]. Following this logic and knowing that EVs have an important protein content, we hypothesized that the proteomic profile is also influenced by AIV infection and TLR ligand stimulation of chicken tracheal cells. We first aimed to characterize the protein content of EVs after treatment. In this study, we have shown that EVs released from TOCs stimulated with AIV, LPS, or polyI:C have distinct proteomic profiles with key functions in cell signaling and immune responses.

During AIV infection or TLR ligand stimulation, the overall release of EVs is affected in terms of contents rather than the amount. In a previously published paper, it was demonstrated that the type of stimulation does not affect the amount of EVs released form cells. It was demonstrated that the type of stimuli can affect the miRNA contents of EVs [33]. First, we identified 140 DE proteins in EVs released from chicken tracheal cells. We found 111 identified proteins in the ExoCarta and Vesiclepedia databases, whereas 29 proteins were not found in these EV proteome databases. Due to the heterogeneous nature of EVs across species and tissues, novel protein identification is expected [51]. More specifically, we characterized the proteome of EVs released from chicken tracheal cells for the first time; therefore, we expected to identify proteins previously found in EVs, but also proteins that are potentially specific to these types of respiratory EVs in chickens. Furthermore, we demonstrated that the proteomic profile of EVs released from chicken tracheal cells depends on the treatment (AIV, LPS, or polyI:C). This suggests that the EV protein content is influenced by viral infection or TLR ligand stimulation. This also expands and reinforces our conclusions drawn from the results of our previous study on miRNAs content in EVs released from chicken tracheal cells. Taken together, this supports the theory that EVs undergo specific cargo-loading and packaging [52]. Accumulating evidence has shown that EVs are master communicators during infection and may be reflected in EV contents [53,54]. In addition, some recent studies have shown that AIVs can be found in EVs and may play a role in the spread of the virus; however, no AIV proteins were detected in the EVs, suggesting that this strain of AIV is not packaged into EVs released from chicken tracheal cells as a means of viral dissemination [55]. This must be further validated by other methods such as viral RNA detection in EVs.

Venn diagram analysis provided insight into the commonalities and specific differences between our treatment groups. The 12 up-regulated proteins and 17 down-regulated proteins common to all the treatment groups may indicate similar roles in immunity or host-pathogen interactions as all the treatments that interact with the immune system. In contrast, the 9, 17, and 4 uniquely up-regulated proteins and 9, 22, and 20 uniquely down-regulated proteins in the AIV, LPS, and polyI:C groups, respectively, suggest that these proteins may be reflective of specific signaling involved by the different stimulations. In addition, protein-protein interaction analysis revealed that proteins within these treatment groups are interconnected. As previously noted, several proteins were up-regulated in one group and down-regulated in another, further supporting the claim that the EV protein contents are highly dependent on the treatment. Uncovering the differences between the treatment groups allowed us to understand the differences in EV protein contents, but to gain a better understanding about the biological purposes of these proteins, we extracted gene ontology terms to predict the functions of the proteomic cargo. We must acknowledge the limitation of having a limited functionally annotated chicken protein database and have integrated the use of the human gene symbols for our functional analysis. This type of analysis is a first step towards deciphering the EV proteins’ specific functions. We found that the terms associated with the proteins in the EVs were greatly varied. We were interested in terms associated with cell signaling and immune responses.

For example, the protein EOMES (R4GH67) was down-regulated in all treatment groups and associated with the biological process term “immune system process”. EOMES (R4GH67) or Eomesodermin is a T-box transcription factor involved in the early developmental process of mesoderm specification during gastrulation [56]. This protein also has roles in the functions of effector and memory T cells, and high levels of Eomesodermin have been associated with CD8+ T cell exhaustion [57,58]. CD8+ T cells are essential for protective immunity against intracellular pathogens, but during chronic infections, constant exposure to antigen or inflammatory signals leads to the deterioration of T cell function [59]. A possible explanation for down-regulation of EOMES (R4GH67) in all treatment groups is a negative regulation to prevent CD8+ T cell exhaustion upon activation of the immune system. Another example is the protein associated with the inflammation mediated by chemokine and cytokine signaling pathways, PLCE1 (A0A1D5PQ57) or Phospholipase C epsilon 1 (down-regulated in the AIV and polyI:C groups) and RGS14 (A0A1D5PPP1). A study in esophageal squamous cell carcinoma showed that inflammation or immune-related TLR4, IL-8, IL-6, and chemokine (C-X-C motif) ligand 2 (CXCL2) were increased upon PLCE1 suppression [60]. This suggests that the downregulation of PLCE1 in the AIV and polyI:C groups may serve to have an upregulation of these the goal is to have an upregulation of these immune immune-related molecules. Finally, up-regulated proteins COL4A1 (A0A1D5P8P3) and COL27A1 (F1NHH4) and down-regulated protein COL1A2 (A0A5H1ZRJ7) from the LPS group were found to be associated with the integrin signaling pathways. EVs express integrins on their surfaces and serve many purposes, such as communication by guiding EVs to specific tissues or cells [61]. Collagens (COL) are members of the integrin family and protein support to different tissues, but also regulate cell growth and differentiation [62]. The presence of several differentially regulated collagen proteins within EVs highlights the fine-tuned regulation of processes and cargo-loading of specific proteins. These associated terms recapitulate the nature of EVs in the immune response and cell signaling. 

EVs can impact the immune response in a variety of different ways, such as through the NF-κB signaling pathway [27,28]. Chicken tracheal epithelial cells mount antiviral responses through similar pathways [32]. These types of signaling pathways require highly regulated communication between cells, highlighting a potential role for EVs as intercellular mediators. AIV infection or TLR ligand stimulation leads to important immunological changes in chicken tracheal cells. Specifically, there is induction in the expression of pro-inflammatory cytokines, interferons, and interferon-stimulated genes [32]. EVs released from tracheal cells also undergo important changes in content under similar stimulation conditions. The presence of proteins involved in immune responses and cell signaling in these EVs indicates a potential correlation between cellular and reflected EV changes. Although this study provides an important overview of the contents of EVs released from chicken tracheal cells under different conditions, i.e., AIV infection, timing is of extreme importance in the context of antiviral responses, and so a key limitation of this study is that it does not evaluate the change in released EV content over time. Investigation into these changes would be complementary to the results presented in this study, as there may be important changes at different time points post-stimulation. Furthermore, although knowledge about these specific EV proteins within provides insight into their potential roles, the specific functions in the context of EVs and the antiviral response would need to be further validated with functional studies evaluating individual proteins.

Knowing that the EVs contained proteins important in immune response and cell signaling, the second aim of this study was to evaluate the potential impact or specific roles that EVs have on macrophage functions or the potential roles in communication. We chose to evaluate the impact of EVs on macrophage function because we previously demonstrated the communication and interaction between chicken tracheal cells and chicken macrophages in the context of antiviral responses [29,30,31,32]. We hypothesized that EVs may play a role here. We demonstrated that EVs released from TOCs impact macrophage function. We first demonstrated that chicken macrophages can uptake EVs rather than simply sense them. Many cells have been shown to uptake EVs, including macrophages [45]. EV populations can be very heterogenous, especially among different species and cell and tissue types, resulting in a variety of potential routes for sensing or uptake [63,64]. In order to determine the exact mechanism for EV uptake, such as endocytosis or phagocytosis, further studies are required. Furthermore, certain specific proteins are required for certain receptor-mediated internalization mechanisms; therefore, a cross-reference to the identified EV proteins would be required.

Following EV uptake, we investigated macrophage activation through evaluation of NO production. NO production in macrophages was evaluated as it is an indicator of macrophage activation [65,66]. In addition, LPS was used as a positive control of macrophage activation as it was previously shown to induce NO production in chicken macrophages [29]. EVs alone (high dose of EV LPS, EV polyI:C, and EV CTRL) were able to induce significant increases in NO production compared to the untreated group, indicating the ability to activate macrophages. Furthermore, we observed a synergistic effect in the EV + LPS groups, with the low dose of EVs inducing a higher amount of NO. This showed that EVs were able to boost the LPS activation of macrophages. This may be due to the existence of negative feedback loops in the NO production pathway [67]. No notable comparisons between EV groups were observed, indicating that the induction of NO production may be due to the presence of EVs themselves and not particularly the EV contents, which we showed to be distinct among the treatment groups. Upon investigating the ability of EV stimulation to impact the phagocytic abilities of macrophages, as phagocytosis is an important function of activated avian macrophages, we found that the only the high dose of EV LPS had a significant impact on phagocytosis compared to the untreated control group [5]. Knowing that LPS is a potent activator of chicken macrophages, these phagocytosis data suggest that EVs may contain specific molecules and components of signaling pathways, which may act in transferring the activation “message” induced by LPS stimulation [29]. Further investigation is needed to determine the exact molecular mechanisms.

The coordination of antiviral responses is mediated through signaling pathways involving many components. Increased relative gene expression of IFN-α for some EV groups at 3 h post-stimulation indicates an early activation of the IFN pathways, which is crucial in the antiviral response [68]. At 18 h post-stimulation, the change in relative expression for both IFN-α and IFN-β is significantly down-regulated for some groups and insignificant for others. This may indicate that the IFN expression is more important at the initial stages of macrophage stimulation. IL-1β, an important mediator of the inflammatory response, was significantly up-regulated for all the groups and the relative gene expression was dose dependent. More specifically, the high dose of EVs inducing higher levels of IL-1β than the low dose of EVs. Furthermore, IL-1β can initiate NO synthesis; therefore, the induced NO production we observed in the macrophages may be due to the EVs themselves or by the increased levels of IL-1β [69]. Finally, both the ISGs RNA-activated protein kinase R (PKR) and melanoma differentiation-associated gene 5 (MDA5) are significantly down-regulated in some EV groups at 18 h post-stimulation. This may indicate a less important role for these molecules at 18 h versus 3 h post-stimulation [70,71]. Taken together, NO production, phagocytosis, and gene expression data for macrophages support the notion that responses to viral infections are complex and, more likely than not, it is a combination or network of proteins and other EV contents that play a role in the induction of the antiviral state and the impact on other cells.

## 5. Conclusions

In conclusion, we investigated the overall protein profile of EVs released from chicken tracheal cells in response to AIV infection and TLR ligand stimulation. We showed that EV contents are influenced by the treatment received by the chicken tracheal cells and that these EVs are enriched in proteins involved in immune responses and cell signaling. Furthermore, we evaluated the impact of EV stimulation on other cells of the immune system and showed that EVs have the ability to activate macrophages. Additional functional studies are required to elucidate the mechanisms responsible for selective EV cargo loading and for the impact of EV stimulation on the antiviral activity and activation of macrophages. Although further functional studies are required to validate specific EV protein functions, this study revealed the role of respiratory EVs in the induction and modulation of antiviral responses against viral infections. A greater understanding of EV contents and functions will ultimately lead to the development of specifically tailored EV therapeutics applicable in the context of infectious viral disease.

## Figures and Tables

**Figure 1 membranes-12-00053-f001:**
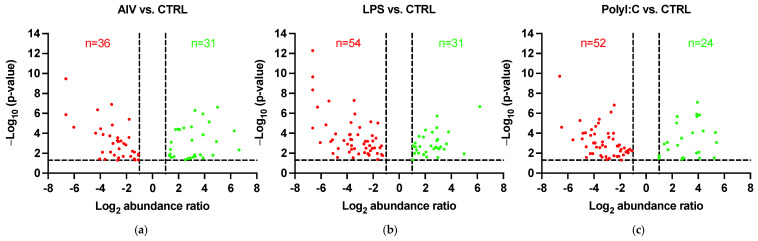
Volcano plots showing DE proteins of EVs from TOCs treated with (**a**) AIV, (**b**) LPS, and (**c**) polyI:C. The horizontal dotted line represents the *p*-value < 0.05 threshold. Up-regulated proteins are represented by green data points and down-regulated proteins are represented by red data points. The vertical dotted lines represent the abundance ratio ≥ 2-fold change threshold. Lists of the up- and down-regulated proteins for AIV, LPS, and polyI:C treatment groups are shown in Table 2 and Table 3, respectively.

**Figure 2 membranes-12-00053-f002:**
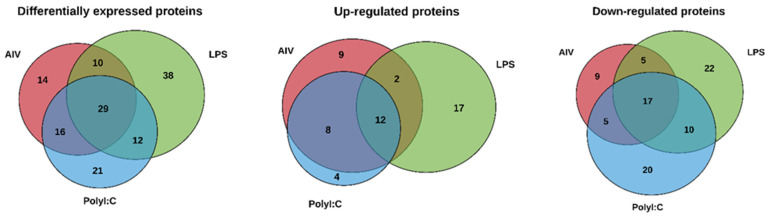
Venn diagram showing DE proteins of EVs from TOCs treated with (**a**) AIV, (**b**) LPS, and (**c**) polyI:C. Lists of the up- and down-regulated proteins for AIV, LPS, and polyI:C treatment groups and intersections of groups are shown in Appendix A.

**Figure 3 membranes-12-00053-f003:**
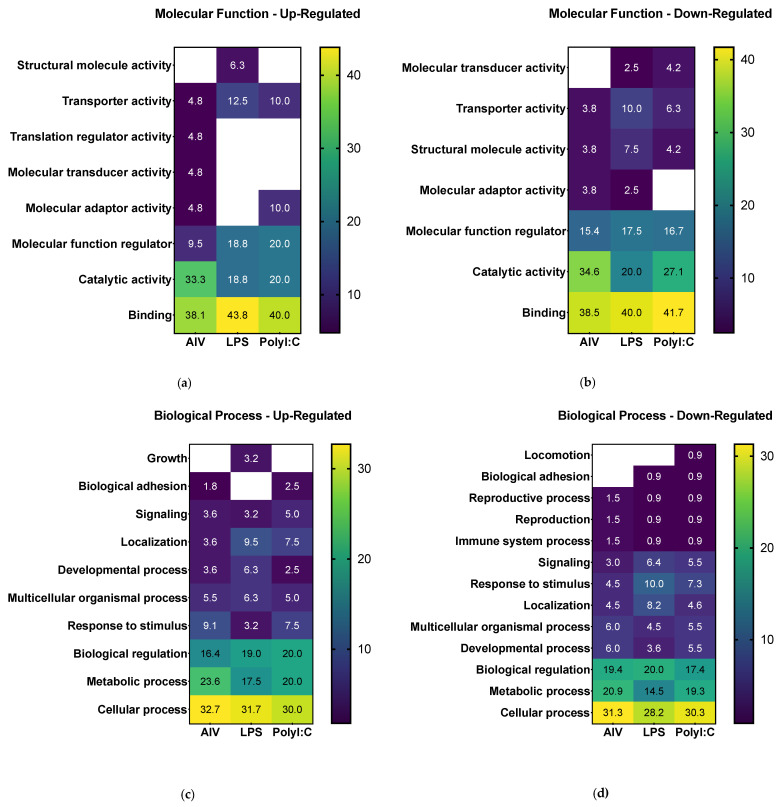
Gene set enrichment analysis of DE proteins of EVs from TOCs treated with AIV, LPS, and polyI:C. Separated by up-regulated and down-regulated proteins, the functional annotation for molecular function (**a**,**b**), biological process (**c**,**d**), cellular component (**e**,**f**), and protein class (**g**,**h**), respectively, were obtained from the PANTHER classification system database. The color intensities represent the percent gene hits against the total number of hits for each term. Detailed lists of the up- and down-regulated proteins and associated gene ontology terms for AIV, LPS, and polyI:C treatment groups are found in Appendix A.

**Figure 4 membranes-12-00053-f004:**
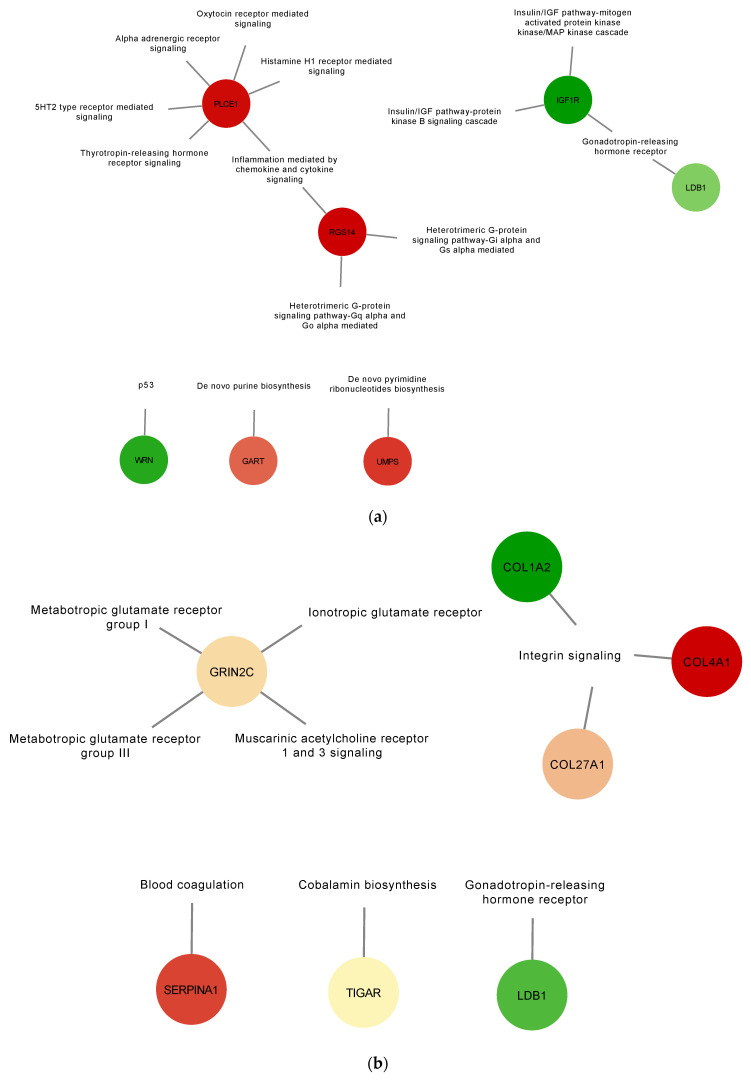
Gene set enrichment analysis of up-regulated and down-regulated proteins of EVs from TOCs treated with AIV (**a**), LPS (**b**), and polyI:C (**c**). Functional annotation for selected pathways was obtained from the PANTHER classification system database. Pathway enrichment results are shown as a network with nodes colored based on relative abundance (compared to EV control treatment group). Selected up-regulated proteins are shown in green, whereas down-regulated proteins are shown in red. The complete lists of up- and down-regulated proteins for AIV, LPS, and polyI:C treatment groups are shown in Appendix A.

**Figure 5 membranes-12-00053-f005:**
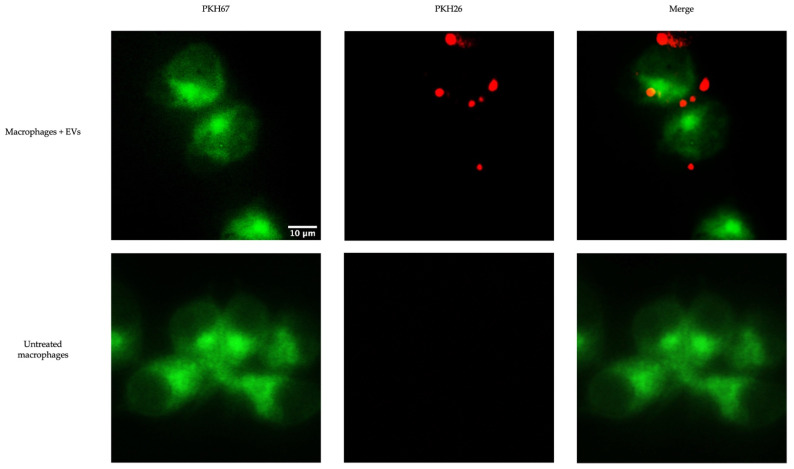
Uptake of EVs by chicken macrophage. PKH67-labelled (**green**) chicken macrophages were treated with 10 μg of the PKH26-labelled (**red**) EVs and incubated at 40 °C and 50% CO_2_ for 2 h. The uptake of EVs by macrophages was detected by fluorescence microscopy.

**Figure 6 membranes-12-00053-f006:**
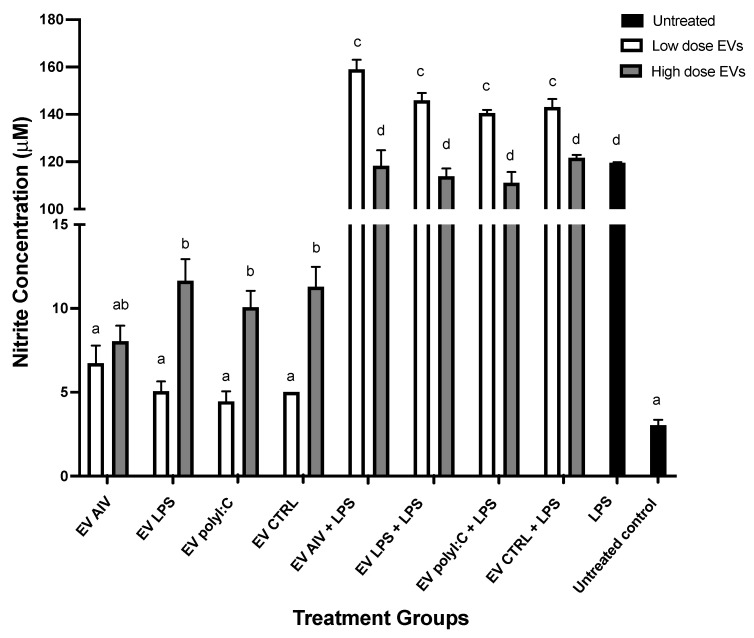
NO production by chicken macrophages stimulated with EVs. Macrophages were treated with either a low (5 µg/mL) or high (50 µg/mL) dose of EVs, with or without LPS treatment. NO production was assessed by the Griess assay. Significant differences (*p*-value < 0.05) are denoted by letters. Groups that are significantly different are represented by different letters. Groups with the same letters are not significantly different. The error bars represent the standard error of mean (SEM).

**Figure 7 membranes-12-00053-f007:**
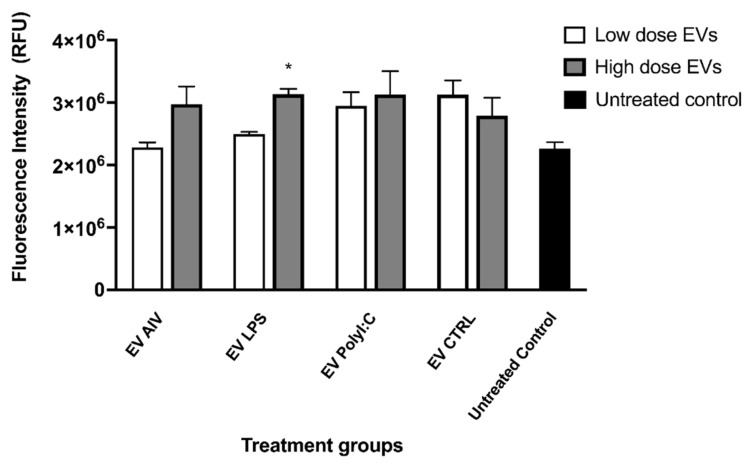
Phagocytosis by chicken macrophage stimulated with EVs. Two doses of EVs were used to treat macrophages, low (5 µg/mL) and high (25 µg/mL). Phagocytosis was assessed by a fluorescent bead-based assay using pHrodo Red *Escherichia coli* Bioparticles Conjugates. No significant differences in fluorescence intensity (relative fluorescence units, RFU) were observed as a result of EV treatment. Treatment groups with significant differences (*p*-value < 0.05) compared to the untreated control group are represented by *. The error bars represent the standard error of mean (SEM).

**Figure 8 membranes-12-00053-f008:**
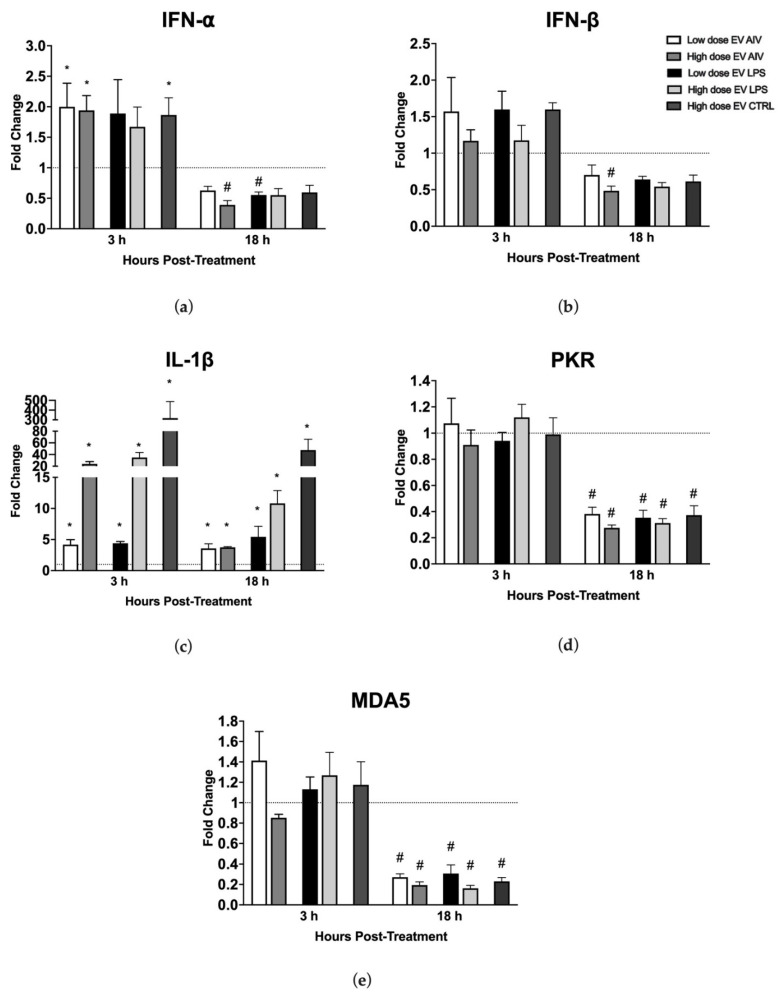
Relative gene expression of chicken macrophages stimulated with EVs. Macrophages were treated with a low dose (5 µg/mL) or a high dose (25 µg/mL) of EVs and collected at two different timepoints, 3 h and 18 h post-stimulation. The relative gene expression of IFN-α (**a**), IFN-β (**b**), IL-1β (**c**), PKR (**d**), and MDA5 (**e**) was measured by RT-qPCR. Treatment groups with significant up-regulation (*p*-value < 0.05) are represented by *, whereas treatment groups with significant downregulation (*p*-value < 0.05) are represented by #. The error bars represent the standard error of mean (SEM).

**Table 1 membranes-12-00053-t001:** Primer sequences used for RT-PCR.

Target Gene	Primer Sequence	Annealing Temperature (°C)	Reference
β-actin	F: 5′-CAACACAGTGCTGTCTGGTGGTA-3′	60	[49]
	R: 5′-ATCGTACTCCTGCTTGCTGATCC-3′		
IFN-α	F: 5′-ATCCTGCTGCTCACGCTCCTTCT-3′	64	[49]
	R: 5′-GGTGTTGCTGGTGTCCAGGATG-3′		
IFN-β	F: 5′-GCCTCCAGCTCCTTCAGAATAC G-3′	64	[50]
	R: 5′-CTGGATCTGGTTGAGGAGGCTGT-3′		
IL-1β	F: 5′-GTGAGGCTCAACATTGCGCTGTA-3′	64	[49]
	R: 5′-TGTCCAGGCGGTAGAAGATGAAG-3′		
MDA5	F: 5′-TGGTACAGGCGTTGGTAAGAG-3′	60	[30]
	R: 5′-GAGCACATCCGCAGGTAGAG-3′		
PKR	F: 5′-GCAAAACCAGCACTGAATGGG-3′	60	[30]
	R: 5′-CGTAAATGCTGTTCCACTAACGG-3′		

**Table 2 membranes-12-00053-t002:** Up-regulated proteins of EVs from TOCs treated with AIV, LPS, and polyI:C. Following differential expression filtering (abundance ratio ≥ 2-fold change and *p*-value < 0.05), 31, 31, and 24 proteins were found to be up-regulated in the AIV, LPS, and polyI:C treatment groups, respectively.

Treatment Group	Accession	Protein Name	Gene Symbol	Abundance Ratio	Log_2_ Abundance Ratio	*p*-Value
**AIV** **(31 proteins)**	A0A1D5PJC0	Uncharacterized protein	-	100.00	6.64	4.57 × 10^−3^
	A0A3Q2TWB1	Phosphoinositide 5-phosphatase	INPP5B	77.51	6.28	5.62 × 10^−5^
	A0A1D5P5K6	PHD-type domain-containing protein	TCF20	32.00	5.00	2.39 × 10^−7^
	A0A3Q2U363	Uncharacterized protein	ADGRA1	30.12	4.91	6.68 × 10^−4^
	E1BU50	Uncharacterized protein	LOC419409	24.96	4.64	1.52 × 10^−2^
	A0A1D5P4T1	Uncharacterized protein	SYTL2	20.43	4.35	7.06 × 10^−6^
	A0A3Q2UIP4	AT-rich interactive domain-containing protein 5B	ARID5B	14.84	3.89	1.36 × 10^−4^
	E1BQH9	Uncharacterized protein	UGGT2	14.84	3.89	2.92 × 10^−2^
	F1P186	Aa_trans domain-containing protein	SLC38A7	14.67	3.87	1.15 × 10^−6^
	A0A1D5P8Q3	Uncharacterized protein	PAPLN	12.78	3.68	1.42 × 10^−2^
	A0A1D5PVH7	Tyrosine-protein kinase receptor	IGF1R	9.73	3.28	2.14 × 10^−2^
	F1NZ61	ZnMc domain-containing protein	MMP27	9.59	3.26	5.12 × 10^−7^
	A0A1L1RVC2	hSH3 domain-containing protein	C8H1orf168	9.47	3.24	1.37 × 10^−2^
	E1BSS2	Ubiquitin specific peptidase 53	USP53	9.46	3.24	2.00 × 10^−2^
	A0A1L1RJ96	FSA_C domain-containing protein	KIAA1109	8.67	3.12	2.45 × 10^−2^
	A0A3Q2UBZ4	Uncharacterized protein	MSLNL	8.18	3.03	4.95 × 10^−4^
	F1NG87	TPR_REGION domain-containing protein	TTC28	8.01	3.00	2.24 × 10^−5^
	F1NX10	Uncharacterized protein	FRAS1	7.57	2.92	2.99 × 10^−2^
	F1NCE1	Eukaryotic translation initiation factor 3 subunit D	EIF3D	6.95	2.80	3.51 × 10^−2^
	A0A1L1RQF9	DNA helicase	WRN	6.67	2.74	3.19 × 10^−2^
	A0A1L1RIR0	FHA domain-containing protein	MCRS1	5.72	2.52	4.38 × 10^−2^
	A0A1D5PHD3	C2H2-type domain-containing protein	ZNF318	5.30	2.41	2.86 × 10^−5^
	Q2PC93	SCO-spondin	SSPO	4.26	2.09	4.20 × 10^−5^
	A0A3Q3A731	Uncharacterized protein	-	3.86	1.95	3.86 × 10^−5^
	E1C1W0	Uncharacterized protein	TRERF1	3.42	1.77	4.11 × 10^−5^
	A0A3Q2UAW2	Rho-GAP domain-containing protein	SYDE1	3.16	1.66	2.17 × 10^−2^
	O42252	LIM domain-binding protein 1	LDB1	2.71	1.44	7.69 × 10^−4^
	E1BQX8	GRIP domain-containing protein	GOLGA1	2.71	1.44	2.59 × 10^−2^
	A0A3Q2U7G6	AF-4_C domain-containing protein	AFF1	2.61	1.38	4.13 × 10^−3^
	R4GGB8	Uncharacterized protein	GAS2L3	2.57	1.36	1.22 × 10^−2^
	A0A1D5PIM5	CARMIL_C domain-containing protein	CARMIL3	2.42	1.28	1.66 × 10^−2^
**LPS** **(31 proteins)**	A0A1D5P5K6	PHD-type domain-containing protein	TCF20	73.16	6.19	2.15 × 10^−7^
	E1C4T7	Uncharacterized protein	SETX	31.65	4.98	1.18 × 10^−2^
	A0A1D5P4T1	Uncharacterized protein	SYTL2	13.80	3.79	7.10 × 10^−5^
	A0A3Q2UHE4	SH3 domain-containing protein	CTTN	12.36	3.63	1.15 × 10^−3^
	A0A3Q2U363	Uncharacterized protein	ADGRA1	10.95	3.45	4.03 × 10^−3^
	A0A1D5P268	Uncharacterized protein	SHROOM4	9.10	3.19	2.28 × 10^−3^
	A0A1D5PFS7	SCA7 domain-containing protein	ATXN7	8.36	3.06	3.63 × 10^−3^
	F1NF87	Exocyst complex component 5	EXOC5	7.78	2.96	4.17 × 10^−4^
	A0A3Q2U9L9	Uncharacterized protein	TRAPPC8	7.77	2.96	2.80 × 10^−2^
	F1NZ61	ZnMc domain-containing protein	MMP27	7.54	2.91	1.84 × 10^−6^
	A0A3Q2UIP4	AT-rich interactive domain-containing protein 5B	ARID5B	7.49	2.90	2.93 × 10^−3^
	A0A1D5NXL6	Uncharacterized protein	PHF3	7.39	2.89	5.47 × 10^−4^
	F1P186	Aa_trans domain-containing protein	SLC38A7	7.35	2.88	2.75 × 10^−5^
	A0A5H1ZRJ7	Collagen alpha-2(I) chain	COL1A2	6.99	2.81	2.57 × 10^−3^
	A0A1D5P7D0	Uncharacterized protein	NUMA1	6.31	2.66	3.58 × 10^−3^
	F1NG87	TPR_REGION domain-containing protein	TTC28	5.84	2.55	7.78 × 10^−5^
	A0A1D5P111	Uncharacterized protein	KCNK5	5.43	2.44	1.89 × 10^−3^
	A0A3Q2UAW2	Rho-GAP domain-containing protein	SYDE1	4.32	2.11	2.35 × 10^−2^
	A0A3Q2UPH9	Dystrophin	DMD	4.17	2.06	7.72 × 10^−3^
	Q2PC93	SCO-spondin	SSPO	4.09	2.03	4.79 × 10^−5^
	A0A3Q2U741	UDENN domain-containing protein	ST5	3.67	1.88	1.35 × 10^−2^
	A0A1D5PHD3	C2H2-type domain-containing protein	ZNF318	3.63	1.86	2.38 × 10^−4^
	A0A1L1RZA6	Uncharacterized protein	LOC420370; ZNF746	3.00	1.59	2.13 × 10^−3^
	E1C1W0	Uncharacterized protein	TRERF1	2.86	1.52	2.03 × 10^−4^
	A0A3Q2UAU4	Uncharacterized protein	TMOD4	2.80	1.49	9.82 × 10^−3^
	O42252	LIM domain-binding protein 1	LDB1	2.54	1.34	4.79 × 10^−3^
	A0A3Q3A731	Uncharacterized protein	-	2.37	1.24	2.27 × 10^−3^
	A0A3Q2TYZ1	Uncharacterized protein	MPDZ	2.34	1.23	9.71 × 10^−4^
	E1C371	Uncharacterized protein	LATS1	2.27	1.18	1.06 × 10^−2^
	F1NCA2	Glycerol-3-phosphate dehydrogenase	GPD2	2.12	1.08	1.93 × 10^−3^
	A0A3Q2U7G6	AF-4_C domain-containing protein	AFF1	2.09	1.06	4.64 × 10^−2^
**PolyI:C (24 proteins)**	A0A3Q2U363	Uncharacterized protein	ADGRA1	42.37	5.40	8.29 × 10^−4^
	R4GGB8	Uncharacterized protein	GAS2L3	39.94	5.32	8.45 × 10^−5^
	A0A1D5PJC0	Uncharacterized protein	-	37.61	5.23	2.98 × 10^−2^
	E1BYS6	Protein kinase C	PKN2	20.82	4.38	5.94 × 10^−5^
	A0A1D5P5K6	PHD-type domain-containing protein	TCF20	17.10	4.10	1.49 × 10^−6^
	E1BU50	Uncharacterized protein	LOC419409	17.00	4.09	3.21 × 10^−2^
	A0A1D5PJI9	Uncharacterized protein	VPS13A	16.70	4.06	7.82 × 10^−3^
	F1NZ61	ZnMc domain-containing protein	MMP27	15.32	3.94	7.70 × 10^−8^
	F1NG87	TPR_REGION domain-containing protein	TTC28	15.03	3.91	2.05 × 10^−6^
	F1P186	Aa_trans domain-containing protein	SLC38A7	14.81	3.89	1.24 × 10^−6^
	E1C1H9	PLD phosphodiesterase domain-containing protein	PLD5	14.58	3.87	9.52 × 10^−3^
	E1BQX8	GRIP domain-containing protein	GOLGA1	11.87	3.57	9.02 × 10^−5^
	E1BSS2	Ubiquitin specific peptidase 53	USP53	7.44	2.90	3.05 × 10^−2^
	A0A1L1RVC2	hSH3 domain-containing protein	C8H1orf168	6.94	2.80	2.96 × 10^−2^
	A0A1L1RJ96	FSA_C domain-containing protein	KIAA1109	6.72	2.75	3.96 × 10^−2^
	A0A1D5P4T1	Uncharacterized protein	SYTL2	6.60	2.72	3.09 × 10^−4^
	A0A3Q2UBZ4	Uncharacterized protein	MSLNL	5.35	2.42	1.61 × 10^−3^
	E1C1W0	Uncharacterized protein	TRERF1	5.23	2.39	2.09 × 10^−6^
	A0A3Q3A731	Uncharacterized protein	-	5.08	2.35	9.16 × 10^−6^
	A0A1D5P3H2	Guanylate cyclase	LOC107055115	3.20	1.68	4.59 × 10^−3^
	A0A1D5PHD3	C2H2-type domain-containing protein	ZNF318	3.09	1.63	8.23 × 10^−4^
	O42252	LIM domain-binding protein 1	LDB1	2.63	1.39	1.18 × 10^−3^
	Q2PC93	SCO-spondin	SSPO	2.01	1.00	1.81 × 10^−2^
	A0A3Q2U7G6	AF-4_C domain-containing protein	AFF1	2.00	1.00	3.24 × 10^−2^

**Table 3 membranes-12-00053-t003:** Down-regulated proteins of EVs from TOCs treated with AIV, LPS, and polyI:C. Following differential expression filtering (abundance ratio ≥ 2-fold change and *p*-value < 0.05), 36, 54, and 52 proteins were found to be down-regulated in the AIV, LPS, and polyI:C treatment groups, respectively.

Treatment Group	Accession	Protein Name	Gene Symbol	Abundance Ratio	Log_2_ Abundance Ratio	*p*-Value
**AIV** **(36 proteins)**	R4GH67	T-box_assoc domain-containing protein	EOMES	0.01	−6.64	3.38 × 10^−10^
	A0A1D5PVY2	Uncharacterized protein	EHMT1	0.01	−6.64	1.35 × 10^−6^
	E1C1D1	Annexin	ANXA7	0.02	−6.03	2.43 × 10^−5^
	E1C5B4	Enhancer of polycomb homolog	EPC1	0.05	−4.36	9.67 × 10^−5^
	F1NJM6	Uncharacterized protein	LOC100857368; ZCCHC6	0.05	−4.21	4.31 × 10^−7^
	A0A3Q2TTI8	Zinc finger protein 644	ZNF644	0.06	−4.05	3.77 × 10^−2^
	A0A1D5PPP1	Uncharacterized protein	RGS14	0.06	−3.98	3.40 × 10^−5^
	A0A1D5PQ57	Phosphoinositide phospholipase C	PLCE1	0.07	−3.84	7.81 × 10^−3^
	E1BU62	DOP1 leucine zipper like protein B		0.07	−3.78	1.33 × 10^−4^
	F1NY55	LEM domain-containing protein	ANKLE2	0.08	−3.63	4.01 × 10^−2^
	E1C8W5	Uncharacterized protein	CHST15	0.1	−3.27	1.75 × 10^−4^
	F1NPH9	OMPdecase	UMPS	0.11	−3.21	7.57 × 10^−3^
	E1C8S3	S1 motif domain-containing protein	SRBD1	0.11	−3.13	1.27 × 10^−7^
	A0A1D5PJ72	Capping protein inhibiting regulator of actin dynamics		0.12	−3.08	1.46 × 10^−5^
	A0A1D5P7I8	Zinc finger protein 516	ZNF516	0.12	−3.03	1.01 × 10^−3^
	A0A3Q2U3A1	Uncharacterized protein		0.13	−2.92	1.53 × 10^−2^
	A0A3Q3ANH5	Uncharacterized protein		0.16	−2.69	5.39 × 10^−4^
	E1BQG1	Uncharacterized protein	TNRC6B	0.16	−2.66	2.82 × 10^−4^
	A0A1D5P251	Uncharacterized protein	MAP1S	0.16	−2.66	4.85 × 10^−2^
	F1NF87	Exocyst complex component 5	EXOC5	0.16	−2.62	6.70 × 10^−4^
	A0A3Q2UHA1	Uncharacterized protein	FAAP100	0.17	−2.59	2.83 × 10^−2^
	A0A1L1RX59	Diadenosine tetraphosphate synthetase	GARS	0.17	−2.56	3.52 × 10^−3^
	R4GGE1	Uncharacterized protein	SATB1	0.19	−2.41	5.86 × 10^−4^
	A0A3Q3A6T8	Uncharacterized protein	ALMS1	0.21	−2.24	2.06 × 10^−2^
	A0A1D5PPH7	UnbV_ASPIC domain-containing protein	CRTAC1	0.22	−2.2	1.03 × 10^−3^
	A0A1D5NU15	Uncharacterized protein	MAGI1	0.24	−2.04	1.43 × 10^−3^
	F1NEI8	Poly(A)-specific ribonuclease PARN	PARN	0.29	−1.8	2.59 × 10^−4^
	E1BYA8	Uncharacterized protein	ERCC6	0.29	−1.78	3.96 × 10^−6^
	A0A1D5P3H2	Guanylate cyclase	LOC107055115	0.31	−1.67	2.09 × 10^−2^
	A0A3Q2TSU2	HECT-type E3 ubiquitin transferase	NEDD4L	0.35	−1.53	6.02 × 10^−3^
	A0A1D5P0N4	Transcriptional activator Myb	MYB	0.39	−1.38	7.61 × 10^−3^
	A0A3Q2U624	Uncharacterized protein	WDR62	0.39	−1.38	3.58 × 10^−2^
	A0A1D5P124	Uncharacterized protein	ANK2	0.39	−1.34	4.80 × 10^−2^
	A0A1D5PCT6	Kinesin-like protein	KIF23	0.47	−1.08	1.56 × 10^−2^
	H9KYN7	Peptidase S1 domain-containing protein	LOC431235; CTRB2; LOC100859877; CTRB1	0.47	−1.08	4.57 × 10^−2^
	F1NWT3	F-box domain-containing protein	FBXO5	0.48	−1.06	1.20 × 10^−2^
**LPS** **(54 proteins)**	A0A1D5NUX8	Uncharacterized protein	SLC25A10	0.01	−6.64	5.15 × 10^−13^
	A0A1D5P8P3	Collagen IV NC1 domain-containing protein	COL4A1	0.01	−6.64	2.19 × 10^−10^
	R4GH67	T-box_assoc domain-containing protein	EOMES	0.01	−6.64	4.54 × 10^−9^
	A0A1D5PCP1	Uncharacterized protein	ERICH3	0.01	−6.64	3.02 × 10^−5^
	F1NN69	Beta-1,4-N-acetylgalactosaminyltransferase	B4GALNT3	0.01	−6.26	2.38 × 10^−7^
	A0A3Q2TZW2	Transcription initiation factor TFIID subunit	RHOGL	0.02	−6.06	8.70 × 10^−4^
	A0A1D5NZF5	BHLH domain-containing protein	USF3	0.02	−5.62	1.45 × 10^−5^
	E1C7T1	SERPIN domain-containing protein	SERPINA1; SPIA1	0.02	−5.39	5.88 × 10^−8^
	A0A3Q2UIH4	Zyxin	ZYX	0.03	−5.25	6.78 × 10^−4^
	F1NT94	Histone acetyltransferase	KAT6A	0.03	−5.12	4.61 × 10^−4^
	F1NY55	LEM domain-containing protein	ANKLE2	0.03	−5.07	1.07 × 10^−2^
	F1NLF0	Uncharacterized protein	EPS15	0.03	−4.88	1.16 × 10^−4^
	E1BU88	Treslin_N domain-containing protein	C10H15orf42; TICRR	0.04	−4.73	2.73 × 10^−2^
	R4GLP0	Cytochrome c oxidase polypeptide VIIc	COX7C	0.04	−4.67	3.42 × 10^−3^
	E1BU62	DOP1 leucine zipper like protein B		0.05	−4.24	5.30 × 10^−4^
	E1BTE7	AAA domain-containing protein	TOR3A	0.06	−3.96	1.21 × 10^−3^
	F1NV58	Uncharacterized protein	SPTBN5	0.07	−3.83	7.92 × 10^−3^
	A0A3Q2U9U5	Ig-like domain-containing protein	ILDR2	0.07	−3.81	1.45 × 10^−4^
	A0A1D5NW78	Uncharacterized protein	SPAG17	0.07	−3.76	1.30 × 10^−4^
	A0A3Q3A6T8	Uncharacterized protein	ALMS1	0.08	−3.73	4.11 × 10^−4^
	A0A1D5P0N4	Transcriptional activator Myb	MYB	0.08	−3.69	8.29 × 10^−6^
	E1C309	Uncharacterized protein	SLC25A19	0.08	−3.66	1.76 × 10^−3^
	P20740	Ovostatin	LOC396151; OVST	0.08	−3.66	4.13× 10^−3^
	E1C1D1	Annexin	ANXA7	0.08	−3.64	4.26 × 10^−4^
	A0A1D5PVP9	Protogenin	PRTG	0.09	−3.53	2.84 × 10^−2^
	E1C8S3	S1 motif domain-containing protein	SRBD1	0.09	−3.45	5.11 × 10^−8^
	F1NDM4	Origin recognition complex subunit 2	ORC2	0.09	−3.43	3.49 × 10^−3^
	F1NJM6	Uncharacterized protein	LOC100857368; ZCCHC6	0.1	−3.4	1.16 × 10^−6^
	F1NHH4	Fibrillar collagen NC1 domain-containing protein	COL27A1	0.11	−3.17	1.39 × 10^−4^
	E1BQP5	Uncharacterized protein	WDR72	0.12	−3.11	1.17 × 10^−2^
	A0A3Q2TSU2	HECT-type E3 ubiquitin transferase	NEDD4L	0.12	−3.03	2.64 × 10^−5^
	R4GGE1	Uncharacterized protein	SATB1	0.15	−2.71	1.19 × 10^−3^
	A0A1D5PVY2	Uncharacterized protein	EHMT1	0.16	−2.65	8.64 × 10^−3^
	E1C8W5	Uncharacterized protein	CHST15	0.17	−2.6	6.38 × 10^−4^
	R4GFN5	Uncharacterized protein	GRIN2C; LOC431090	0.17	−2.55	3.10 × 10^−3^
	F1NWT3	F-box domain-containing protein	FBXO5	0.19	−2.41	6.47 × 10^−4^
	A0A1D5P124	Uncharacterized protein	ANK2	0.19	−2.4	1.12 × 10^−2^
	E1C5B4	Enhancer of polycomb homolog	EPC1	0.2	−2.33	1.26 × 10^−2^
	A0A3Q2U888	Rhomboid domain-containing protein	RHBDF2	0.2	−2.31	1.61 × 10^−2^
	P08287	Histone H1.11L	HIST1H111L; HIST1H1C	0.21	−2.26	1.73 × 10^−3^
	A0A3Q2U624	Uncharacterized protein	WDR62	0.22	−2.21	5.68 × 10^−3^
	A0A3Q2TXE9	Uncharacterized protein		0.22	−2.17	7.11 × 10^−6^
	E1BQG1	Uncharacterized protein	TNRC6B	0.24	−2.07	3.35 × 10^−3^
	R4GIZ6	TIGAR	C1H12orf5; TIGAR	0.24	−2.06	1.72 × 10^−4^
	H9KYN7	Peptidase S1 domain-containing protein	LOC431235; CTRB2; LOC100859877; CTRB1	0.25	−2.02	6.02 × 10^−4^
	A0A1D5PPH7	UnbV_ASPIC domain-containing protein	CRTAC1	0.27	−1.89	2.97 × 10^−3^
	A0A3Q3AGH4	AIG1-type G domain-containing protein	GIMAP7L5	0.29	−1.78	3.10 × 10^−4^
	E1C908	Uncharacterized protein	FYB	0.31	−1.68	1.07 × 10^−2^
	A0A3Q3ANH5	Uncharacterized protein		0.32	−1.64	4.30 × 10^−2^
	E1BYA8	Uncharacterized protein	ERCC6	0.32	−1.63	9.60 × 10^−6^
	A0A1D5PE26	Uncharacterized protein	DENND4A	0.36	−1.47	2.83 × 10^−3^
	F1NEI8	Poly(A)-specific ribonuclease PARN	PARN	0.39	−1.36	1.86 × 10^−3^
	A0A1L1RKD5	Uncharacterized protein	WISP1	0.39	−1.36	1.33 × 10^−2^
	Q5F393	Nuclear receptor coactivator	NCOA1	0.41	−1.27	1.70 × 10^−2^
**PolyI:C** **(52 proteins)**	R4GH67	T-box_assoc domain-containing protein	EOMES	0.01	−6.64	1.93 × 10^−10^
	E1BU62	DOP1 leucine zipper like protein B		0.01	−6.48	2.51 × 10^−5^
	E1C309	Uncharacterized protein	SLC25A19	0.02	−5.61	4.58 × 10^−4^
	A0A1D5NY78	SCD domain-containing protein	STAG1	0.03	−5.07	5.22 × 10^−6^
	A0A3Q2UFS2	AIP3 domain-containing protein	SRCIN1	0.03	−4.9	3.68 × 10^−4^
	A0A1D5PES4	Receptor protein-tyrosine kinase	EPHA5	0.04	−4.85	1.01 × 10^−4^
	A0A1D5PVY2	Uncharacterized protein	EHMT1	0.04	−4.65	9.19 × 10^−5^
	A0A1D5PQ57	Phosphoinositide phospholipase C	PLCE1	0.04	−4.6	2.26 × 10^−3^
	A0A3Q3A9V6	LIM zinc-binding domain-containing protein	ZNF185L	0.04	−4.58	2.73 × 10^−2^
	E1C264	Uncharacterized protein	CDH9	0.05	−4.28	1.09 × 10^−2^
	E1C1D1	Annexin	ANXA7	0.05	−4.25	1.62 × 10^−4^
	A0A3Q2U9U5	Ig-like domain-containing protein	ILDR2	0.06	−4.19	4.27 × 10^−5^
	A0A1D5NVF3	Sorting nexin-17	SNX17	0.06	−4.09	1.12 × 10^−2^
	A0A1D5P8D2	JmjC domain-containing protein	JMJD1C	0.06	−4.03	9.26 × 10^−4^
	F1NLF0	Uncharacterized protein	EPS15	0.07	−3.95	1.18 × 10^−4^
	A0A5H1ZRJ7	Collagen alpha-2(I) chain	COL1A2	0.07	−3.92	9.29 × 10^−4^
	A0A3Q2TSU2	HECT-type E3 ubiquitin transferase	NEDD4L	0.07	−3.87	2.01 × 10^−5^
	E1BQP5	Uncharacterized protein	WDR72	0.07	−3.81	2.45 × 10^−3^
	F1NCA2	Glycerol-3-phosphate dehydrogenase	GPD2	0.08	−3.66	9.44 × 10^−6^
	A0A1D5PJ72	Capping protein inhibiting regulator of actin dynamics		0.08	−3.6	3.91 × 10^−6^
	E1BQF4	TGF-beta receptor type-2	AMHR2	0.09	−3.55	2.75 × 10^−3^
	A0A3Q2TU08	Uncharacterized protein	CDC5L	0.09	−3.48	1.71 × 10^−2^
	F1NV58	Uncharacterized protein	SPTBN5	0.1	−3.38	2.55 × 10^−2^
	E1BTE7	AAA domain-containing protein	TOR3A	0.1	−3.35	7.38 × 10^−4^
	A0A3Q2U624	Uncharacterized protein	WDR62	0.1	−3.35	1.55 × 10^−3^
	E1C5B4	Enhancer of polycomb homolog	EPC1	0.12	−3.12	2.24 × 10^−3^
	A0A1D5P1A1	Uncharacterized protein	C5H11ORF9	0.12	−3.08	2.26 × 10^−2^
	A0A3Q2UHA1	Uncharacterized protein	FAAP100	0.12	−3.03	4.32 × 10^−2^
	R4GGE1	Uncharacterized protein	SATB1	0.13	−2.91	1.04 × 10^−4^
	P08287	Histone H1.11L	HIST1H111L; HIST1H1C	0.14	−2.86	2.58 × 10^−4^
	A0A3Q2UG75	Ras-responsive element-binding protein 1	RREB1	0.14	−2.82	3.39 × 10^−2^
	E1C8S3	S1 motif domain-containing protein	SRBD1	0.16	−2.69	7.57 × 10^−7^
	A0A3Q3ANH5	Uncharacterized protein		0.17	−2.53	3.83 × 10^−4^
	F1NWT3	F-box domain-containing protein	FBXO5	0.18	−2.5	1.02 × 10^−4^
	A0A3Q2U0V9	Uncharacterized protein	SLC11A2	0.18	−2.47	1.82 × 10^−2^
	E1BYA8	Uncharacterized protein	ERCC6	0.18	−2.45	1.51 × 10^−7^
	A0A1D5P0N4	Transcriptional activator Myb	MYB	0.22	−2.22	1.91 × 10^−3^
	A0A3Q3A6T8	Uncharacterized protein	ALMS1	0.22	−2.22	2.09 × 10^−2^
	A0A1D5PD16	Uncharacterized protein	DNAH10	0.23	−2.14	2.01 × 10^−3^
	A0A1L1RX59	Diadenosine tetraphosphate synthetase	GARS	0.24	−2.09	2.08 × 10^−2^
	A0A1D5PBZ9	Protein-tyrosine-phosphatase	PTPRB	0.24	−2.05	1.48 × 10^−3^
	F1P5A5	28S ribosomal protein S31, mitochondrial	MRPS31	0.26	−1.97	6.18 × 10^−3^
	E1BQG1	Uncharacterized protein	TNRC6B	0.26	−1.92	4.09 × 10^−3^
	A0A3Q2U888	Rhomboid domain-containing protein	RHBDF2	0.27	−1.9	4.90 × 10^−2^
	A0A1D5P7I8	Zinc finger protein 516	ZNF516	0.3	−1.74	1.50 × 10^−2^
	H9KYN7	Peptidase S1 domain-containing protein	LOC431235; CTRB2; LOC100859877; CTRB1	0.31	−1.67	3.00 × 10^−3^
	Q5F393	Nuclear receptor coactivator	NCOA1	0.35	−1.54	7.34 × 10^−3^
	A0A1D5PGG8	Nucleolar complex protein 3 homolog	NOC3L	0.36	−1.48	6.74 × 10^−3^
	A0A3Q2U9J3	Ubiquitinyl hydrolase 1	OTUD7A	0.38	−1.38	9.29 × 10^−3^
	F1NQ24	Uncharacterized protein	DENND4C	0.39	−1.37	5.08 × 10^−3^
	F1NEI8	Poly(A)-specific ribonuclease PARN	PARN	0.42	−1.25	4.00 × 10^−3^
	A0A3Q2TXE9	Uncharacterized protein		0.47	−1.09	5.90 × 10^−3^

## Data Availability

Data are contained within the article or Appendix A.

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
