# Peer review of "Characterization of the Role of Extracellular Vesicles Released from Chicken Tracheal Cells in the Antiviral Responses against Avian Influenza Virus"

_membranes, 2021, doi:10.3390/membranes12010053_

Round 1

Reviewer 1 Report

The manuscript from O’Dowd, et al., investigates the protein profile and related signalling pathways in chicken tracheal cells originated extracellular vesicles during the avian influenza infection. The study shows that the extracellular vesicles originating from tracheal cells can influence the macrophages and overall immune response. The study is unique since there are not many studies evaluating extracellular vesicles (EVs) and changes in its content during an inflammatory response. On the other hand, a functional approach giving mechanistic insights are lacking.

Major points:

  • Mechanisms leading to triggered changes in tracheal cells during the avian influenza virus infection and how this translates to change in EV content? A more detailed discussion in this regard would make this study more comprehensive.
  • Is the overall release of EVs from the tracheal cells affected during the avian influenza virus infection? The authors probably have insights already during EV extraction in this regard?
  • Does the amount of EVs released per time change during an avian influenza virus infection?
  • Question 1-4 would ascertain if there are more mechanisms that are altered in the tracheal cells apart from the EV content as the authors describe. How would these factors influence the EV content released? How are the signalling pathways that authors mention influenced in this regard?

Minor points:

  • Table 2 could be represented as series of plots for easy readability.
  • Figure 1, data points are too small to read.
  • Images in Figure 5 lack contrast.
  • Images with better resolution in Figure 5 is important.

Author Response

Dear Reviewer 1,

Thank you very much for taking the time to review our manuscript. We would like to thank you for the opportunity to improve our manuscript before its publication.

Below, you will find responses addressing each point brought up in the review report. In addition, we have highlighted the changes within the manuscript (where applicable).

Firstly, we have done a final read-through to identify and fix any English language mistakes we may have missed. We have taken into consideration the comments and suggestions in order to improve the research design, results, and conclusions.

1) A functional approach giving mechanistic insights are lacking.

Response: thank you for this comment. Through the revisions made based on the reviewers’ comments and suggestions, we hope to provide an improved explanation of our functional approach regarding EVs and the antiviral response against AIV. This manuscript aimed to provide an overview of the contents of EVs released from chicken tracheal cells under different conditions, namely, AIV infection. In this way, we provide potential targets or proteins of interest for future functional studies. We acknowledge that this message must be clear, and so we have modified the concluding paragraph as follows (line 778-783):

Original: Ultimately, this study revealed the role of respiratory EVs in the induction and modulation of antiviral responses against viral infections and a greater understanding of EV contents and functions will lead to the development of specifically tailored EV therapeutics applicable in the context of infectious viral disease.

Modified: While, as aforementioned, further functional studies are required to validate specific EV protein functions, this study revealed the role of respiratory EVs in the induction and modulation of antiviral responses against viral infections. A greater understanding of EV contents and functions will ultimately lead to the development of specifically tailored EV therapeutics applicable in the context of infectious viral disease.

We have planed to continue on this work. We will focus on the function of identified proteins in the context of viral infections in future studies.

Major points:

2) Mechanisms leading to triggered changes in tracheal cells during the avian influenza virus infection and how this translates to change in EV content? A more detailed discussion in this regard would make this study more comprehensive.

Response: thank you for this comment. We have modified the discussion section to address this point and provide more information on the triggered changes in tracheal cells during AIV infection and how it potentially translates to EV content. The following sentences were added (line 697-703):

AIV infection or TLR ligand stimulation leads to important immunological changes in chicken tracheal cells. Namely, there is induction in the expression of pro-inflammatory cytokines, interferons, and interferon-stimulated genes [32]. EVs released from tracheal cells also undergo important changes in content under similar stimulation conditions. The presence of proteins involved in immune responses and cell signaling in these EVs indicates a potential correlation between cellular and reflected EV changes.

3) Is the overall release of EVs from the tracheal cells affected during the avian influenza virus infection? The authors probably have insights already during EV extraction in this regard?

Response: thank you for this question. This is an excellent point to address in the discussion. As a result, we have added the following sentences to the discussion (line 622-626):

During AIV infection or TLR ligand stimulation, the overall release of EVs is affected in terms of contents rather than the amount. In the previously published paper, it has been demonstrated that the type of stimulation does not affect the amount of EVs released from cells. It has been demonstrated that the type of stimuli can affect the miRNA contents of EVs [33]. In our first study evaluating EV miRNA contents, we optimized the EV isolation conditions and protocol. We did not observe any significant differences between the amount of EVs released under stimulation or infection conditions and the untreated control conditions [33].

4) Does the amount of EVs released per time change during an avian influenza virus infection? Question 1-4 would ascertain if there are more mechanisms that are altered in the tracheal cells apart from the EV content as the authors describe.

Response: thank you for this question. We did not evaluate the EV release from tracheal cells as a function of time. In our first study evaluating EV miRNA contents (O’Dowd et al. 2020), we chose the 24 h timepoint post-stimulation based on other EV studies and the studies using TLR ligand and AIV to stimulate chicken tracheal cells. While this would be an interesting aspect to investigate, this goes beyond the scope of our study.

We have added a couple of sentences in the discussion to address this limitation (line 703-706):

Although this study provides an important overview of the contents of EVs released from chicken tracheal cells under different conditions, namely, AIV infection, timing is of extreme importance in the context of antiviral responses, and so a key limitation of this study is that it does not evaluate the change in released EV content over time. Investigation into these changes would be complementary to the results presented in this study, as there may be important changes at different time points post-stimulation.

5) How would these factors influence the EV content released?

Response: Factors such as the changes occurring in the tracheal cells themselves and the different time points would definitely greatly impact the EV contents and resulting functions. However, in this study, we focus on the overview of EV contents under specific conditions, and we can only speculate about other experimental conditions. We mention that there are limitations to choosing to focus on only one condition and state that further investigation in this regard is needed. We hope that the changes made based on your previous suggestions help clarify this aspect of the manuscript.

6) How are the signalling pathways that authors mention influenced in this regard?

Response: Although we make an effort to tie in the information we have about chicken tracheal cells with the results from our EVs studies, for the purpose of this study, we chose to focus on the EV contents and functions specifically. We agree that this aspect is still very important to address and therefore, we have added this short section in the manuscript discussion (line 693-697):

EVs can impact the immune response in a variety of different ways, such as through the NF-κB signaling pathway [27,28]. Chicken tracheal epithelial cells mount antiviral responses through similar pathways [32]. These types of signaling pathways require highly regulated communication between cells, highlighting a potential role for EVs as intercellular mediators.

Minor points:

7) Table 2 could be represented as series of plots for easy readability.

Response: thank you for this suggestion. As we wanted to include all details for the proteins, we chose to have a detailed table in the main text. Figure 1 (line 429) shows a summary of up- and down-regulated proteins for each group and acts as a graphical representation or summary of this table. However, after evaluation of our figure legend, we agree that this is not explicitly clear. We have therefore modified the figure legend to clarify, and we believe that this clarification replaces the need for a series of plots (line 430-433).

Original: Figure 1. Volcano plots showing DE proteins of EVs from TOCs treated with (a) AIV, (b) LPS and (c) polyI:C. The horizontal dotted line represents the p-value < 0.05 threshold. The vertical dotted lines represent the abundance ratio ≥ 2-fold change threshold. Lists of the up- and down-regulated proteins for AIV, LPS and polyI:C treatment groups are shown in Tables 2 and 3, respectively.

Modified: Figure 1. Volcano plots showing DE proteins of EVs from TOCs treated with (a) AIV, (b) LPS and (c) polyI:C. The horizontal dotted line represents the p-value < 0.05 threshold. Up-regulated proteins are represented by green data points and down-regulated proteins are represented by red data points. The vertical dotted lines represent the abundance ratio ≥ 2-fold change threshold. Lists of the up- and down-regulated proteins for AIV, LPS and polyI:C treatment groups are shown in Tables 2 and 3, respectively.

8) Figure 1, data points are too small to read.

 Response: We have enlarged the figure panels within the manuscript rather than enlarge the data points as enlarging the data points resulted in very crowded graphs and the overlapping of the data points. This helped improve the readability of the graph. Please see the preview of figures below to compare change in size (line 429).

9) Images in Figure 5 lack contrast. Images with better resolution in Figure 5 is important.

 Response: thank you for these suggestions. We have edited the figure to improve the contrast and resolution. This has helped us attain the maximum quality for our figure based on the original quality of the fluorescence microscopy images. Please see the preview of figures below to compare change in contrast/resolution (line 539).

Finally, we would like to thank you once again for the incredibly insightful and valuable comments and suggestions contained within the review report.

Reviewer 2 Report

In this paper, O'Dowd and co-workers characterized the protein profile of EVs during AIV infection, thus evaluating the impact of their stimulation on chicken macrophages. 140 differentially expressed proteins were identified and were shown, via proteomics bio-informatic tools, to be involved in immune responses and cell signaling pathways. In addition, macrophages activation by EVs was also demonstrated.

I found the paper well written, the methods described throughtly and the amount of data satisfactory. Thus I suggest publshing it after MINOR REVISIONS.

1) TABLE S2: Intersecting sets of proteins

I think it would be beneficial to report in this table the log2 abundance ratios and the relative p-values for each condition so that, as an example, looking at row 8 it is immediatly clear that the protein O42252 has given ratios in the AIV, LPS & PolyI:C samples, and that these ratios have a given associated confidence. I know that something similar is done in the main text table 2, but I think this would benefit the straightforwardness of the data.

2) Section 3.2: Proteins Found in EVs Released from TOCs Have Functions in Cell Signaling and Immune 
System Processes

At line 501 the authors mention they have exploited STRING to build protein-protein interaction networks with the aim of "illustrating and increasing the overall understanding of the relationships between the identified DE proteins". I think this part is a bit neglected. Could the authors add few lines to better explain the STRING results? Furthermore, in the methods part related to this section I read that a low confidence score was used. Have the authors tried to perform the analysis with at least a medium confidence score (0.4)? I think this should give a better idea on the networks.

3) Figure 6: I find the caption a bit confusing and also the choice of putting a and b on the bars. Could the caption be explained better?

Author Response

Dear Reviewer 2,

Thank you very much for taking the time to review our manuscript. We would like to thank you for the opportunity to improve our manuscript before its publication.

Below, you will find responses addressing each point brought up in the review report. In addition, we have highlighted the changes within the manuscript (where applicable).

Firstly, we have done a final read-through to identify and fix any English language mistakes we may have missed. We have taken into consideration the comments and suggestions in order to improve the research design, results, and conclusions.

1) TABLE S2: Intersecting sets of proteins

I think it would be beneficial to report in this table the log2 abundance ratios and the relative p-values for each condition so that, as an example, looking at row 8 it is immediately clear that the protein O42252 has given ratios in the AIV, LPS & PolyI:C samples, and that these ratios have given associated confidence. I know that something similar is done in the main text table 2, but I think this would benefit the straightforwardness of the data.

Response: thank you for this excellent suggestion to improve the clarity of this table (S2). We have modified Table S2 to include the log2 abundance ratios and associated p-values. This will improve the overall understanding of the intersecting datasets.

2) Section 3.2: Proteins Found in EVs Released from TOCs Have Functions in Cell Signaling and Immune System Processes

At line 501 the authors mention they have exploited STRING to build protein-protein interaction networks with the aim of "illustrating and increasing the overall understanding of the relationships between the identified DE proteins". I think this part is a bit neglected. Could the authors add few lines to better explain the STRING results? Furthermore, in the methods part related to this section, I read that a low confidence score was used. Have the authors tried to perform the analysis with at least a medium confidence score (0.4)? I think this should give a better idea on the networks.

Response: thank you for this suggestion. After reviewing this section, we agree that this aspect of the manuscript is indeed neglected. As a result, we first reviewed the STRING results and performed the analysis using medium confidence. The figure has been updated. Although we see fewer interaction edges, the confidence is increased and, therefore, the information we pull from the networks is more reliable.

We also included a description of the edge colors in the figure legend (line 787-793):

Original: Figure S1. Network analysis of protein-protein interactions among AIV up-regulated (a), AIV down-regulated (b), LPS up-regulated (c), LPS down-regulated (d), polyI:C up-regulated (e) and polyI:C down-regulated (f) groups of EVs from TOCs, obtained from STRING database.

Modified: Figure S1. Network analysis of protein-protein interactions among AIV up-regulated (a), AIV down-regulated (b), LPS up-regulated (c), LPS down-regulated (d), polyI:C up-regulated (e) and polyI:C down-regulated (f) groups of EVs from TOCs, obtained from STRING database, where blue edges represent known interactions from curated databases, purple edges represent experimentally determined known interactions, yellow edges represent interactions from textmining and black edges represent interactions from co-expression.

In addition, we have expanded on the STRING results paragraph to provide a better explanation for this analysis and the results (line 502-511).

Original: Moreover, to illustrate and increase the overall understanding of the relationships between the identified DE proteins, protein-protein interaction networks were built using the STRING database (Figure S1).

Modified: Moreover, to illustrate and increase the overall understanding of the relationships between the identified DE proteins, protein-protein interaction networks were built using the STRING database (Figure S1). The results of this analysis demonstrate a connected network of proteins within the different treatment groups. For example, in the AIV up-regulated treatment group, we found interactions between the proteins SSPO (Q2PC93), ADGRA1/GPR123 (A0A3Q2U363), PAPLN (A0A1D5P8Q3), FRAS1 (F1NX10), and ARID5B (A0A3Q2UIP4). In the LPS down-regulated group, we found interactions between KAT6A (F1NT94), RHOGL (A0A3Q2TZW2), EPC1 (E1C5B4), and EHMT1 (A0A1D5PVY2), as well as between ANK2 (A0A1D5P124) and SPTBN5 (F1NV58).

Finally, in the discussion section, we added the following sentence to clarify the purpose of STRING analysis and to better tie these results into the overall conclusions (line 653-654): In addition, protein-protein interaction analysis revealed that proteins within these treatment groups are interconnected.

3) Figure 6: I find the caption a bit confusing and also the choice of putting a and b on the bars. Could the caption be explained better?

Response: thank you for bringing this point to our attention. We have modified the figure legend to provide a better explanation of the results and the description of significant differences between groups (lines 561-564).

Original: Figure 6. NO production by chicken macrophage stimulated with EVs. Two doses of EVs were used to treat macrophages, low (5 µg/mL) and high (50 µg/mL). NO production was assessed by Griess assay. Treatment groups with significant differences (p-value < 0.05) are represented by different letters, while no significant difference is indicated by the same letter. The error bars represent the standard error of the mean (SEM).

Modified: Figure 6. NO production by chicken macrophages stimulated with EVs. Macrophages were treated with either a low (5 µg/mL) or high (50 µg/mL) dose of EVs, with or without LPS treatment. NO production was assessed by the Griess assay. Significant differences (p-value < 0.05) are denoted by letters. Groups that are significantly different are represented by different letters. Groups with the same letters are not significantly different. The error bars represent the standard error of the mean (SEM).

Furthermore, we chose to use letters to denote significant differences between groups as this was the most effective way to present the data considering the number of treatment groups. We hope that the modification of the figure legend provides a clearer explanation for Figure 6.

Finally, we would like to thank you once again for the incredibly insightful and valuable comments and suggestions contained within the review report.
